

# Multifractal evaluation of simulated precipitation intensities from the COSMO NWP model

Daniel Wolfensberger[1], Auguste Gires[2], Ioulia Tchiguirinskaia[2], Daniel Schertzer[2], and Alexis Berne[1]

[1]LTE, Ecole polytechnique fédérale de Lausanne (EPFL), Lausanne, Switzerland
[2]HMCO, Ecole des Ponts, UPE, Champs-sur-Marne, France

*Correspondence to:* Daniel Wolfensberger (daniel.wolfensberger@epfl.ch)

**Abstract.** The framework of universal multifractals allows to characterize the spatio-temporal variability of fields over a wide range of scales with only a limited number of scale-invariant parameters. In this work, we perform a multifractal analysis of simulated fields of water contents in liquid, solid and gas phase from the COSMO numerical weather prediction model during three different events (one cold front associated with heavy snowfall, one stationary
front with stratiform rain and one summer convection event) over Switzerland. The multifractal parameters of precipitation intensities at the ground are also compared with those obtained from the Swiss radar composite. The results of the analysis show that the COSMO simulations exhibit spatial scaling breaks that are not present in the radar data, indicating that the model is not able to simulate the observed variability at all scales. The impact of the topography on these conclusions was assessed by comparing a very steep area to a mostly flat area. It was observed
that the topography does not seem to play a dominant role in the multifractal characterization of the COSMO water contents. Additionally, a spatio-temporal multifractal analysis of the COSMO simulations and the radar composite was performed and compared with a simplified scaling model of space-time variability.

## 1 Introduction

Validation of precipitation fields simulated by a numerical weather prediction model is a delicate task as reference
data (rain gauges, radar scans) are typically available at a different spatial and temporal resolution than the model. Traditional point-based verification scores are generally unable to provide sufficient information about the forecast quality as they do not consider the spatial structure of the data and are affected by the so-called "double penalty" (Gilleland et al., 2009). Indeed, small displacements in the simulated forecast features will be penalized twice, once for missing the observation and again for giving a false alarm. The impact of this double penalty is related to the
variability of the simulated fields, which tends to increase with the resolution of the model. Numerous methods have been proposed in recent years to address this issue. Some methods rely on the use of traditional scores but applied on filtered fields, estimating the forecast performance as a function of scale and precipitation intensity (e.g., Mittermaier et al., 2013; Ebert, 2008) while others detect specific features on forecast and verification fields and





compare these features based on their attributes (e.g., Davis et al., 2006; Wernli et al., 2008). Other methods rely on the separation of scales with the use of space-frequency methods such as the 2D wavelet transform (Vasić et al., 2007).

Multifractals offer a convenient way to analyze the variability of complex geophysical systems globally over a wide range of scales. In the context of multifractals, the statistical properties of a field are related to the resolution by a power-law (Schertzer and Lovejoy, 1987). Universal Multifractals (UM) are a framework based on the concept of multiplicative cascades, which allows to analyse and simulate a high variability across scales with only a small number of parameters with physical meaning (e.g., Schertzer and Lovejoy, 1987; Lovejoy and Schertzer, 2007). In meteorology, UM have been used to study a large variety of complex natural phenomena such as the distribution of rainfall intensities at the ground (e.g., Marsan et al., 1996; Gires et al., 2015a, b), atmospheric turbulence (e.g., Parisi and Frisch, 1985a; Schertzer and Lovejoy, 2011) or climate change (e.g., Schmitt et al., 1995; Royer et al., 2008).

Gires et al. (2011), used the UM framework to compare simulations of Meso-NH, a non-hydrostatic numerical weather prediction (NWP) model developed by Météo-France, with composite radar images during a heavy convective rainfall event. This comparison showed that both the radar quantitative precipitation estimation (QPE) and the model simulations were generally characterized by similar ranges of scaling and agreed quite well with a simple space-time scaling model.

In the current work we use a similar approach over Switzerland using simulations from the COSMO NWP model and extend the analysis to three different synoptic situations (snowstorm, convective summer precipitation, stratiform rainfall). In addition to the comparison of the properties of the simulated and radar-derived precipitation intensities at the ground, we also perform a UM analysis of liquid water, ice water, water vapour and total water contents at different altitude levels. Ultimately, the goal is to assess the capacity of the model to reproduce the observed variability as well as the expected variability of water contents with altitude. A special emphasis is put on the comparison of the operational one-moment microphysical scheme with a more advanced two-moment scheme. The impact of topography on the multifractal characteristics of the simulated field is studied as well, in order to assess if the observed results can be generalized to other areas.

This article is structured as follows: in section 2 the COSMO model as well as the Swiss radar composite are described briefly. The studied events as well as the radar data sets and model variables are then described in details, followed by a summarized description of the UM framework. In section 3, a spatial and temporal analysis of the different water contents is performed for both microphysical schemes. This analysis is complemented with an evaluation of the influence of topography on the observed trends. In section 4, a similar analysis is conducted for the precipitation intensities on the ground and the results are compared with the UM analysis of the radar composite. Finally section 4 gives a summary of the main results and concludes this work.





## 2  Description of the data

### 2.1  The COSMO model

The COSMO model is a mesoscale limited area numerical weather prediction model initially developed as the Lokal Modell (LM) at the Deutscher Wetterdienst (DWD). It is now operated and developed by various weather services
in Europe, including Switzerland. Besides its operational uses it is also used for scientific purposes in weather prediction and for regional climate simulations. The COSMO model is a non-hydrostatic model based on the fully compressible primitive equations integrated using a split-explicit third-order Runge–Kutta scheme (Wicker and Skamarock, 2002). The spatial discretization is based on a fifth-order upstream advection scheme on an Arakawa C-grid with Lorenz vertical staggering. Height-based Gal-Chen coordinates are used in the vertical (Gal-Chen and
Somerville, 1975). The model uses a rotated coordinate system where the pole is displaced to ensure approximatively horizontal resolution over the model domain. Sub-grid scale processes are taken into account with parametrizations. In particular grid-scale clouds and precipitation are parametrized operationally with a one-moment scheme with five hydrometeor categories: rain, snow, graupel, ice crystals and cloud droplets. Snow is assumed to be in the form of rimed aggregates of ice-crystals that have become large enough to have an appreciable fall velocity. Cloud ice
is assumed to be in the form of small hexagonal plates that are suspended in the air and have no appreciable fall velocity. The particle size distributions (PSD) are assumed to be exponential for all hydrometeors, except for rain where a gamma PSD is assumed:

$$N(D) = N_0 D^\mu \exp\left(-\Lambda \cdot D\right) \quad \mathrm{m^{-3} mm^{-1}} \tag{1}$$

where $D$ is the equivolume diameter, $N_0$ is the intercept parameter ($\mathrm{m^{-3} mm^{-1}}$), $\lambda$ the slope parameter ($\mathrm{mm^{-1}}$)
and $\mu$ the unitless shape parameter

In the one-moment scheme, which is used operationally, the only free parameter of the PSDs are the slope parameters $\lambda$ which can be obtained from the prognostic moment of order three (mass densities). The intercept parameters $N_0$ are either assumed to be constant or in the case of snow to be temperature dependent. The scale parameter $\mu$ is equal to zero (exponential PSDs) for all hydrometeors except for rain where it is set to 0.5 by default.
Mass-diameter relations as well as velocity-diameter relations for the precipitating hydrometeors are assumed to be power-laws.

A more advanced two-moment scheme with a sixth hydrometeor category, hail, was developed for COSMO by Seifert and Beheng (2006). In this scheme all PSDs are assumed to be gamma distributions where the intercept and slope parameters are free parameters that can be obtained from the prognostic moments of order zero (concentration)
and order three (mass fractions). Hydrometeor fall velocities are assumed to be power-laws except for rain where an empirical relation by Rogers et al. (1993) is used. As this scheme significantly increases the overall computation time it is currently not used operationally.





In COSMO, the interaction of various microphysical processes and their feedback on the simulated flow fields are represented by a system of budget equations for $q^x$, the specific mass fraction in $\mathrm{kg}_x$ per $\mathrm{kg}_{\mathrm{air}}$ for hydrometeor $x$.

$$\frac{\partial q^x}{\partial t} + \mathbf{v} \cdot \nabla q^x - \frac{1}{\rho} \frac{\partial P^x}{\partial z} = S^x - \frac{1}{\rho} \nabla \cdot \mathbf{F}^x \tag{2}$$

where $S^x$ represent the microphysical sources and sink per unit mass of moist air, $\mathbf{F}^x$ are the turbulent fluxes and
$P^x$ denotes the precipitation or sedimentation fluxes defined by $P^x = \rho q^x v_T^x$, where $v_T^{(j)}$ is the terminal fall velocity of hydrometeor $j$. The precipitation intensity at the ground is then simply the sum of the sedimentation fluxes of all hydrometeors at the lowest model level. In terms of terminal velocities COSMO assumes power-laws $v_T = aD^b$, where $D$ is the particle equivolume diameter, for all hydrometeor types except for rain in the two-moment scheme, where a slightly more refined formula is used (Seifert and Beheng, 2006).

Numerically this system of differential equations is treated with a time splitting method, in which the advection terms $\mathbf{v} \cdot \nabla q^x$ are first integrated over a COSMO time step (20 sec) and the budget equations are then solved for the microphysical source terms and sedimentation only. In the operational microphysical scheme the source terms include (1) *nucleation and depositional growth of cloud ice*, (2) *autoconversion of cloud water to rain*, (3) *collection mechanisms*, (4) *diffusional growth of rain and snow* and (5) *melting and freezing mechanisms*. Details about the
parameterization of all these source terms can be found in Doms et al. (2011).

In the operational set-up, the COSMO model uses a prognostic turbulent kinetic energy (TKE) closure at level 2.5 for the parametrization of atmospheric turbulence. This scheme is similar to Mellor and Yamada (1982), the main difference being the use of variables that are conserved under moist adiabatic processes: total cloud water and liquid water potential temperature. Additionally, a so-called "circulation term" is included which describes the
transfer of non-turbulent sub-grid kinetic energy from larger-scale circulation toward TKE. The reader is referred to Baldauf et al. (2011) and the model documentation (Doms et al., 2011) for a more in-depth description of the various COSMO sub-grid parametrizations.

## 2.2   Simulated events and model setup

Three different events were simulated, corresponding to typical synoptic situations observed over Switzerland. A
brief description of the events is given in Table 1 and 500 hPa geopotential and temperature charts are shown in Figure 1. To simulate these events, COSMO was used in its version 5.01 with the standard MeteoSwiss operational namelists at a 2 km resolution (0.02° angular resolution), a set-up known operationally as "COSMO-2" (COSMO, 2015), which was used for forecast until beginning of 2016. As was done in similar studies (Bohme et al., 2009), a spin-up time of 12 hours was used to account for the cold start of the model. For the initial and boundary conditions,
analysis forcings of MeteoSwiss obtained with the COSMO-7 model run at 7 km resolution were used in order to run the model in analysis mode by correcting it with the most accurate information available at the time of simulation. In addition, the events were also simulated using the non-operational two-moment scheme, while keeping all other



| Day | Timeline | Nb. of time steps | Description |
|---|---|---|---|
| 26 March 2010 | 08:00 - 18:00 | 144 (12 h) | Crossing of a strong cold front causing sudden drop of temperature followed by heavy graupel and snowfall as well as strong winds |
| 8 April 2014 | 02:00 - 10:00 | 144 (12 h) | Stationary front with widespread stratiform precipitation over Switzerland |
| 13 August 2015 | 12:00 - 24:00 | 144 (12 h) | Strong summer convection triggered by the presence of very warm and wet subtropical air over Switzerland |

**Table 1.** Short description of the three considered precipitation events

namelist parameters unchanged. For all simulations, model outputs were written every 5 minutes of simulated time, which corresponds to the temporal resolution of the Swiss radar composite.

Four different specific water contents were computed from the model outputs: $QV$, the water vapour content, $LWC$, the liquid water content (rain and cloud droplets), $IWC$, the solid (ice) water content (snow, graupel and ice
crystals) and $TWC$, the total water content which is the sum of all previous quantities. All quantities are in g·m$^{-3}$ of air. The different quantities were linearly interpolated from hybrid model levels to fixed altitude levels ranging from 3500 to 10000 m with a step of 250 m. The lower limit of 3500 m was chosen in order to avoid missing values due to topography. Additionally all fields were resized to be square with a size being a power of two. For the comparison of specific water contents, a domain of 256 x 256 km$^2$ centered within the simulation domain and covering most of
Switzerland was chosen (Study zone 1 in Figure 2).

The time series of these water quantities averaged over the study domain are represented in Figure 3 as a function of altitude. For the first event, precipitation is mainly in solid form except close to the ground. A quick increase in ice water content can be observed starting from 09:00 when the system starts to develop over the study area and snowfall becomes more intense. Starting from 13:00, the precipitation intensities are at their maximum and there is
a gradual decrease of all water contents, as the system moves out of the study area. After the cold front crosses the study area (around 15:00), the freezing level height decreases significantly from around 2000 m to 1300 m and the liquid water content drops strongly at altitudes above 3500 m. For the second event (8 April 2014), the situation is quite similar, except that the IWC is located at higher altitude and due to the stability of the atmosphere almost all of the LWC is located below the melting layer at altitudes smaller than 2400 m, which are not taken into account
in the analysis. For the last event (13 August 2015), which is a summer convection event, it can be seen that the LWC extends much higher in altitude and that almost no IWC is visible below the freezing level height (around 4000 m). The IWC and LWC are at their maximum level between 18:00 and 20:00, period during which the most intense convective cells are present.

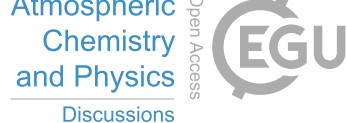



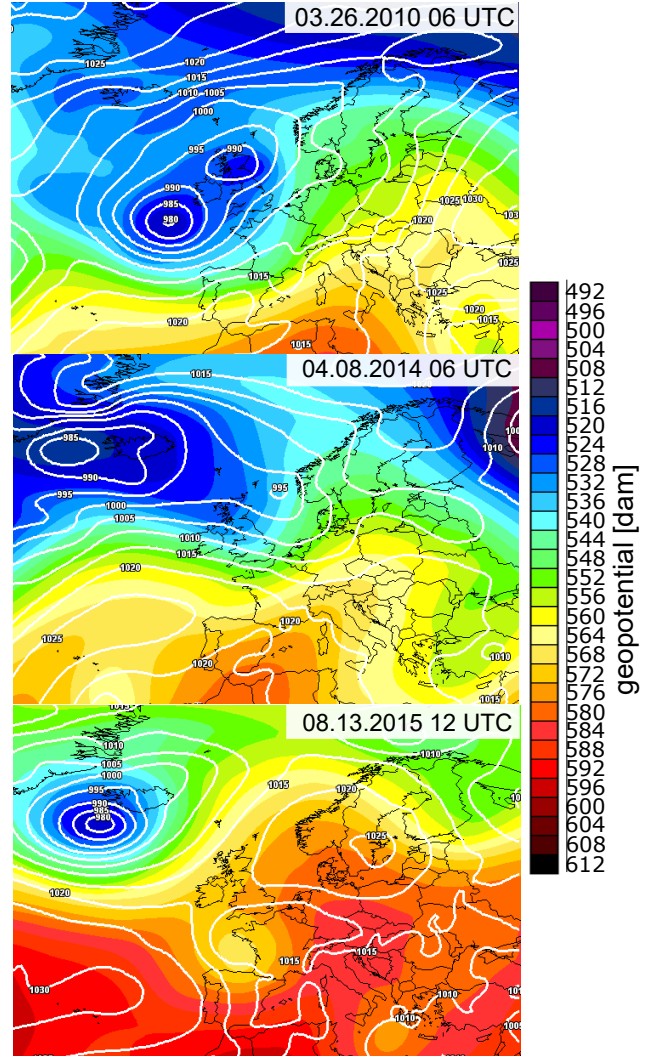

**Figure 1.** 500 hPa geopotentials and PMSL for the three considered events

Figure 4 shows the time series of mean and maximum convective available potential energy (CAPE) during the three events. CAPE is a measure of atmospheric instability. Values larger than 1000 J·kg$^{-1}$ generally indicate a potential for the development of deep convection (Wallace and Hobbs, 2006). The CAPE time series show the great stability of the stratiform rain event (8 April 2014) during the whole studied period. The heavy snowfall event caused by the crossing of a cold front shows higher values of CAPE, due to the greater slope of temperature caused by the cold front. Finally the last event (13 August 2015) is as expected the only one which shows CAPE values large enough for locally strong convection.





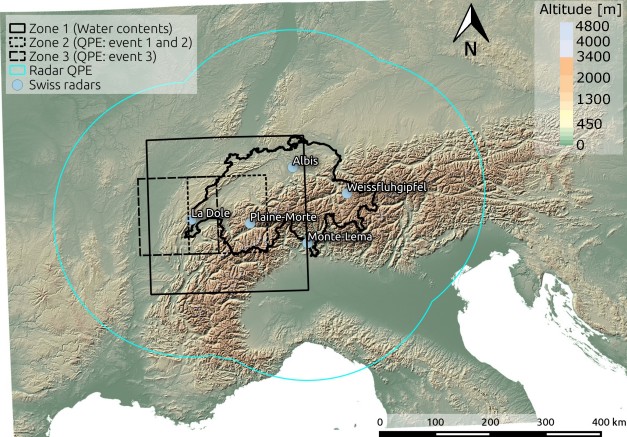

**Figure 2.** Situation map showing the theoretical maximum extent of available QPE (light blue), the Swiss operational radars (blue dots) as well as the region used for the multifractal study of COSMO water contents (zone 1) and the sub-regions centered over the precipitation events used in the QPE analysis (zone 2 and 3)

## 2.3   Radar data

Precipitation intensities at the ground simulated by the COSMO model were compared with the quantitative precipitation estimation (QPE) product from the Swiss operational radar composite. The Swiss radar composite consists of the plane (PPI) measurements of the four [1] operational polarimetric C-band radars. The QPE product of MeteoSwiss is computed in the following way. The linear equivalent radar reflectivity measurement at up to six $1° \times 1° \times 83\text{m}$ clutter-free radar bins, corrected for partial beam-blocking, are averaged to derive polar $1° \times 1° \times 500\text{m}$ radar bins. Reflectivity measurements are then converted to equivalent precipitation intensity with a $Z - R$ relationship. The precipitation estimation at the ground is extrapolated from multi-radar observations aloft using a weighting function that depends on the altitude above the ground and the radar visibility. Corrections for the vertical profile of reflectivity (VPR) is done with an average profile based on aggregation over a few hours and over the visible part of the area located less than 70 km around the radar. More information on the MeteoSwiss QPE estimation can be found in Germann et al. (2006). Note that the Plaine-Morte radar was only installed in 2014 and was thus not available during the first event (26 March 2010). The Swiss radar composite extends radially up to 250 km from every single radar (Figure 2). However, the quality of the product is better closer to the radar and in the areas where the radar scanning domains overlap. To perform a comparison of rain intensities a smaller field of 128x128 km$^2$ was chosen in the center of the domain where the quality of the product is optimal (Zone 1 in Figure 2). For the second event (8 April 2014) the domain was moved slightly to the left in order to better follow the evolution of the precipitation event (Zone 2 in Figure 2)

---

[1]The Weissfluhgipfel radar was not yet installed at the time of the considered events



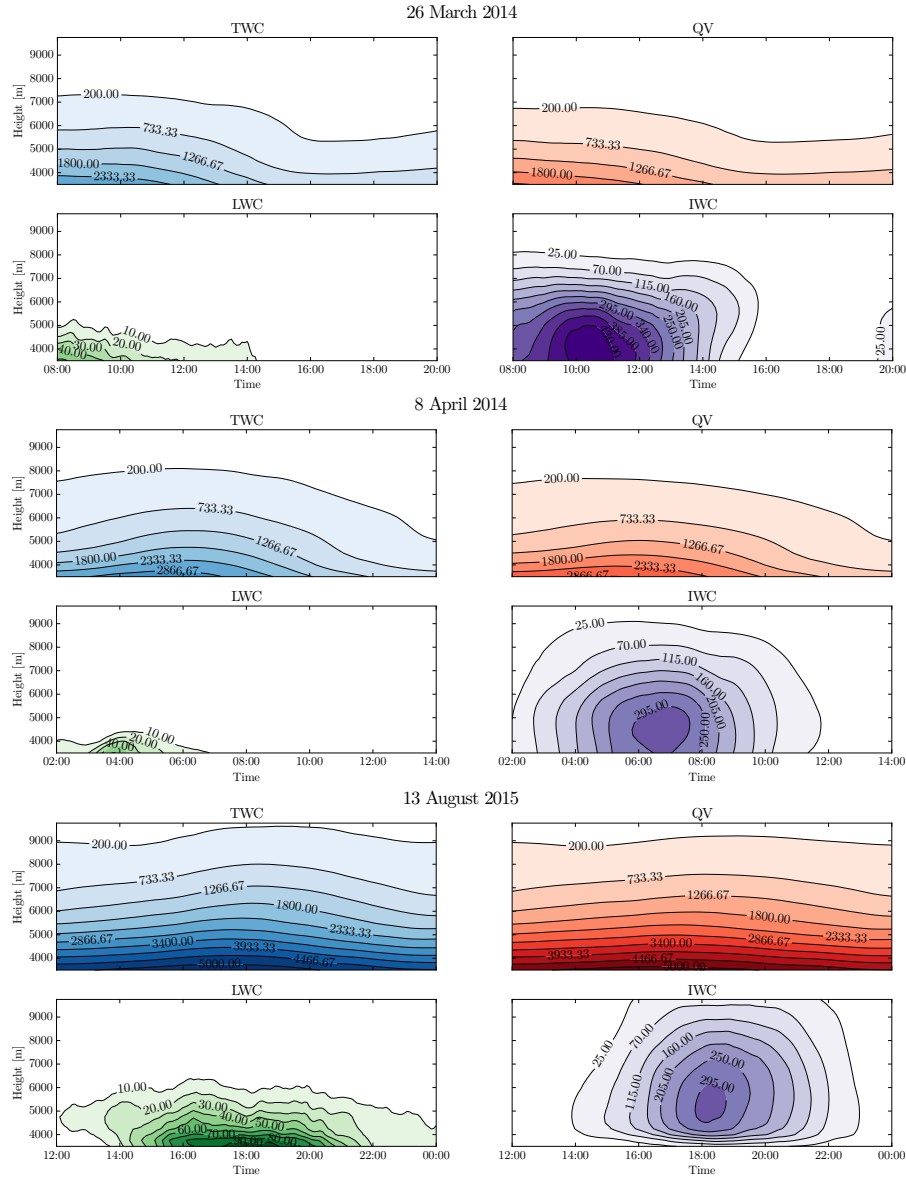

**Figure 3.** Time series of average mass densities as a function of altitude for the four water quantities over Zone 1 in mg m$^{-3}$ during the three events simulated with the one-moment microphysical scheme of COSMO





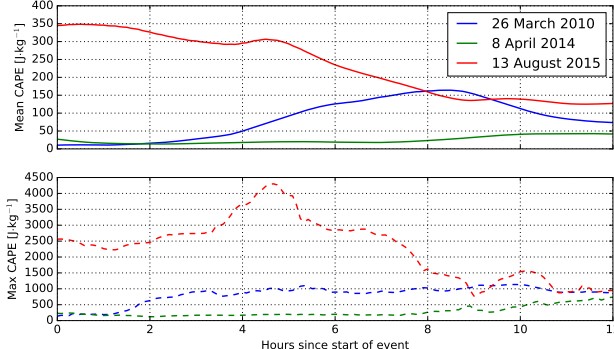

**Figure 4.** Time series of mean (top) and maximum (bottom) convective available potential energy (CAPE) in J·kg$^{-1}$ during the three events. CAPE is a measure of the positive buoyancy of an air parcel and thus an indicator of atmospheric instability

## 3 The UM framework

### 3.1 Multifractality

Let $\epsilon$ be a normalized (divided by its mean) conservative field, which can be one or two dimensional (time serie or spatial map). In the multifractal framework, $\epsilon_\lambda$, the field at resolution $\lambda$ is obtained by up-scaling the field measured or simulated at the maximum resolution to the resolution $\lambda$ which is defined by the ratio between outer scale $L$ and observation scale $l$ ($\lambda = L/l$).

If $\epsilon$ is multifractal, its statistical moments $q$ scale with resolution:

$$\langle \epsilon_\lambda^q \rangle \approx \lambda^{K(q)} \tag{3}$$

Where $K(q)$ is the moment scaling function. For a conservative field $\langle \epsilon_\lambda \rangle = 1$.

It can be shown (Schertzer and Lovejoy, 1987) that this is equivalent to the following relation between probabilities of exceeding a certain threshold :

$$\Pr(\epsilon_\lambda \leq \lambda^\gamma) \approx \lambda^{-c(\gamma)} \tag{4}$$

where $c(\lambda)$ is the co-dimension function which is convex and increasing, $\gamma$ is a so-called singularity, which is independent of scale. $\lambda^\gamma$ can thus be seen as a scale dependent threshold. The functions $K(q)$ and $c(\gamma)$ are related by a Legendre transform (Parisi and Frisch, 1985b).

The quality of this scaling can be studied with the Trace Moment (TM) method which consists for each moment $q$ in a log-log plot of the up-scaled fields as a function of the resolution $\lambda$, the slope being the moment scaling function.



In the universal multifractal framework (Schertzer and Lovejoy, 1987), the moment scaling function $K(q)$ can be fully characterized with only two parameters, $\alpha$ and $C_1$:

$$K(q) = \frac{C_1}{\alpha - 1} (q^\alpha - q) \tag{5}$$

$C_1$ is the mean intermittency co-dimension and measures the clustering of the (average) intensity at increasing scales. $C_1$ is equal to zero when the field is homogeneous. $\alpha$ is the multifractality index and measures the clustering variability with respect to the intensity level, $\alpha \in [0, 2]$.

The size of the sample limits the insight one can get of a statistical process. For multifractal processes, if $N_s$ samples are available this will result in a maximum singularity $\gamma_s$ and moment order $q_s$ beyond which the values of the statistical estimates of the co-dimension and moment scaling function are not considered as reliable ((Schertzer and Lovejoy, 1987), (Lovejoy and Schertzer, 2007)). It can be shown that in the multifractal framework we have:

$$q_s = \left( \frac{D + D_s}{C_1} \right)^{\frac{1}{\alpha}} \quad \text{and} \quad \gamma_s = \alpha' C_1 \left( \frac{D + D_s}{C_1} \right)^{\frac{1}{\alpha'}} \tag{6}$$

where $\frac{1}{\alpha} + \frac{1}{\alpha'} = 1$ and $D_s$ is the sampling dimension defined by $N_s = \lambda^{D_s}$.

Example of the use of $\gamma_s$ can be found in Royer et al. (2008) who investigated the impact of climate change on rainfall extremes using a climate model. They observed an increase of $\gamma_s$ over time which could result in a possible increase in the intensity of rainfall extremes over the next hundred years. Douglas and Barros (2003) and Hubert et al. (1993) also used the maximum singularity $\gamma_s$ in the estimation of probable maximum precipitation.

In order to perform a multifractal analysis the field $\epsilon$ needs to follow the following properties

1. The size $N$ of the field needs to be the same in all dimensions i.e. $\epsilon \in \mathbb{R}^{N^D}$.

2. $N$ needs to be a power of two.

In this work, the UM parameters are estimated with the Double Trace Moment (DTM) method (Lavallée et al., 1993). This method relies on the fact that in the context of UM, the moment scaling function $K(q, \eta)$ of the field $\epsilon_\lambda^{(\eta)}$, obtained by raising the field $\epsilon$ at a power $\eta$ and up-scaled at resolution $\lambda$ can easily be expressed as a function of $\alpha$ (Lavallée et al., 1993):

$$\left\langle \left( \epsilon_\lambda^{(\eta)} \right)^q \right\rangle \approx \lambda^{K(q, \eta)} = \lambda^{\eta^\alpha K(q)} \tag{7}$$

$\alpha$ is thus the slope of the linear part of $K(q, \eta)$ as a function of $\eta$ in a log-log plot.

## 3.2 Non-conservative fields

In the case of a non-conservative field $\phi$, we have $\langle \phi_\lambda \rangle \neq 1$.



One way to consider non-conservative fields within the UM framework it to assume that they can be expressed as:

$$\phi_\lambda = \epsilon_\lambda \lambda^{-H} \tag{8}$$

where $H$ is the non-conservation parameter ($H = 0$ for conservative fields) and $\epsilon$ is a conservative field characterized by a moment scaling function $K_c(q)$ with parameters $C_1$ and $\alpha$.

The moment scaling function of the non-conservative field $\phi_\lambda$ is then given by:

$$K(q) = K_c(q) - Hq \tag{9}$$

$H$ can be related to the spectral slope $\beta$ by:

$$\beta = 1 + 2H - K_c(2) \tag{10}$$

where $\beta$ is the exponent of the power law that characterizes the relation between power spectrum and wave
numbers:

$$E(k) \propto k^{-\beta} \tag{11}$$

Hence the larger the value of the slope $\beta$, the shorter the decorrelation range. If $\beta$ is larger than the dimension of the field, the field is non-conservative.

$\epsilon_\lambda$ can be estimated from $\phi_\lambda$ with a fractional integration (for $H < 0$) or differentiation (for $H > 0$) of order $H$,
which is equivalent to a multiplication by $k^H$ in the Fourier space. In practice however, for $H > 0$, particularly when $H > 0.5$, $\epsilon_{\lambda_{\max}}$ (the field $\epsilon$ at the maximum resolution) is often approximated by the renormalized absolute fluctuations of the field.

$$\epsilon_{\lambda_{\max}}(i) = \frac{|\phi_{\lambda_{\max}}(i+1) - \phi_{\lambda_{\max}}(i)|}{\langle |\phi_{\lambda_{\max}}(i+1) - \phi_{\lambda_{\max}}(i)| \rangle}, \quad i = 1, 2, .., N \tag{12}$$

### 3.3    Spatio-temporal analysis

The multifractal analysis of time series of two-dimensional fields, such as the ones considered in this study, can be performed both in space, by considering an ensemble of two-dimensional fields (one sample for every time step) or in time, by considering an ensemble of one-dimensional time series (one sample for every coordinate in the two-dimensional field).





A simple spatio-temporal scaling model (e.g., Marsan et al., 1996; Deidda, 2000; Macor et al., 2007; Radkevich et al., 2008) is based on the hypothesis of an anisotropy coefficient between space and time:

$$K_{\text{space}}(q) = \frac{K_{\text{time}}(q)}{1 - H_t} \tag{13}$$

where $H_t$ is the anisotropy coefficient between space and time, which in the theory of Kolmogorov (Kolmogorov (1962),Marsan et al. (1996)) is equal to 1/3. This result implies identical $\alpha$ and proportional $C_1$ and $H$ parameters:

$$\frac{C_{1,\text{space}}}{C_{1,time}} = \frac{H_{\text{space}}}{H_{\text{time}}} = \frac{1}{1 - H_t} \tag{14}$$

## 4 Spatial analysis of specific water contents (QV, TWC, LWC and IWC)

### 4.1 Spectral and scaling analysis

A spectral analysis of the COSMO water contents at different altitudes was performed both in space and in time (Section 3.3). It was observed that for both microphysical schemes all COSMO water contents display very good scaling properties in time independently of the altitude. In space however, whereas the total water content (TWC), the water vapour contentration (QV) and the solid water content (IWC) generally scale well even at higher altitudes, the liquid water content (LWC) at high altitudes (5000 m or more) shows values of $\beta$ (inverse of the spectral slope) close to zero at larger observation scales ($> 8$ km) indicating that there is only limited scaling (no straight line of the power spectra) for these scales at these altitudes, which can be explained by the near absence of liquid water. An example is given in Figure 5 for the one-moment scheme and the first event. Finally, for the last (convective) event only, IWC behaves similarly to LWC and shows a scaling break at around 8 km with no apparent scaling ($\beta \approx 0$) at larger scales.

Figure 6 shows the power spectrum slopes $\beta$ and non-conservation parameter $H$ as a function of height for the one-moment scheme of COSMO. Clearly for all events and almost all heights $\beta$ is larger than 2 (the dimension of the field) for IWC, TWC and QV, indicating that these quantities have long decorrelation range and great values of H, corresponding to non conservative fields. In contrast, the liquid water content seems to be a more conservative quantity but as expected does not scale well at altitudes above 5000 m, where it becomes very intermittent. It can also be observed that the relative order of water contents in terms of $\beta$ and $H$ is similar between events, LWC having the lowest values, followed by IWC, TWC and QV. Values of $\beta$ for the two-moment scheme are very similar albeit on average slightly larger (not displayed).

In terms of altitude dependence, a similar trend can be observed for the two first events, with a marked decrease in LWC and smaller variations in IWC, QV and TWC. The third event (convective) however shows an increase in $H$ with altitude which is not observed on the other events.

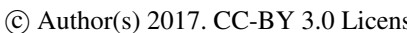



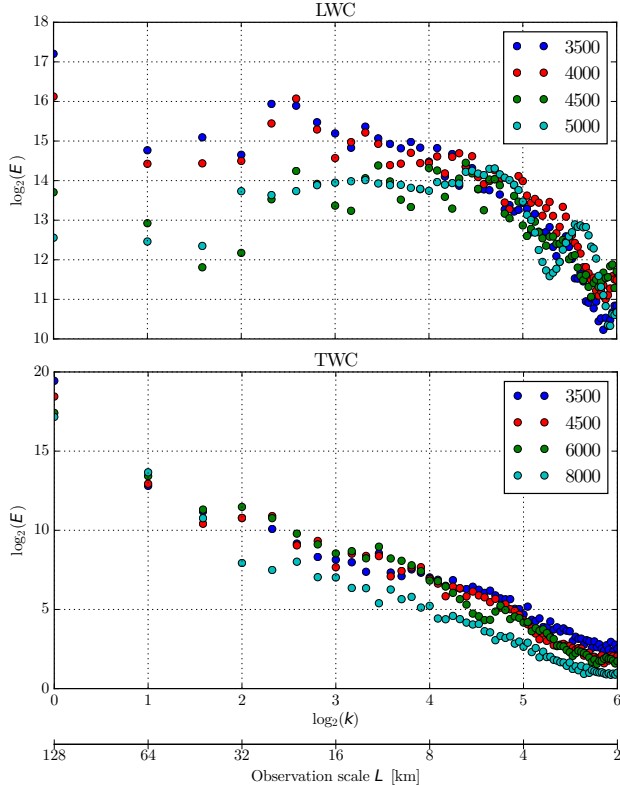

**Figure 5.** Power density spectrum for the LWC (top) and the TWC (bottom) during the first event (26 March 2010), for the one-moment scheme

Since IWC, QV and TWC are strongly non conservative the multifractal analysis for IWC, TWC and QV was performed on the fluctuations of these fields. Taking the fluctuations of the field is a convenient approximation to the fractional differentiation of the field, but in case of large $H$ is generally not sufficient to make the fields truly conservative. Indeed, in our case, the transformed fields (fluctuations) still exhibit $H > 0$ (around 0.1 to 0.4 for QV

5   and TWC and around 0.5 for IWC, with a general increase of $H$ with altitude).

Figure 7 shows the TM analysis for $q = 1.5$ for the ice water content for the COSMO one-moment scheme. It can be observed that the TM curves are not strictly straight lines but show some curvature at increasing observation scale. This trend is particularly visible for the first two events which show a scaling break around 16 km. For the last events, the TM curves are more linear, with however a slight scaling break around 64 km. The TM curves of

10  LWC are quite similar with however a quicker decrease of $R^2$ with altitude, whereas the TM curves of TWC and QV show a high $R^2$ and a single scaling regime at all heights and for all events. In order to properly take into account these scaling breaks in LWC and IWC the further multifractal analysis should be performed separately for all scaling regimes. However in our case, to facilitate the intercomparison of events and water contents, we used one





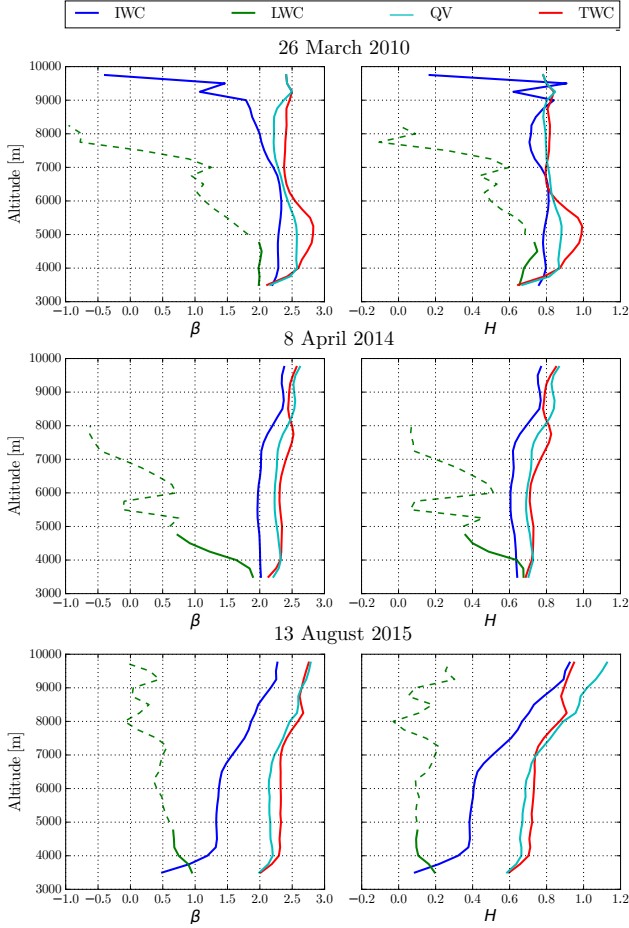

**Figure 6.** Power spectrum slope $\beta$ (left) and non-conservation parameter $H$ (right) as a function of height for the three events and the four water quantities (for the one-moment microphysical scheme only). LWC is generaly characterized by poor scaling properties above 5000 m, hence the corresponding values are shown with a dashed line.

single scaling range (2-256 km) for all events and all water contents. A refined analysis of this issue will of carried out at ground level with comparison with radar data in section 5.1.

## 4.2 Influence of topography

To study the influence of the topography on the non-conservativity of the COSMO water contents, a spectral and
5    TM analysis was performed for the two first events in two different regions of $128 \times 128$ km$^2$. The first region is located in France and centered around point (47.7°N/4.5°E) in quite flat terrain (mean alt. = 295 m, stdev of alt. = 109 m). The second region, which is located in the Swiss Alps and centered around point (46.1°N/7.53°E) is located in very steep terrain (mean alt. = 1714 m , stdev of alt. = 784). Note that the last event (13 August 2015) was not





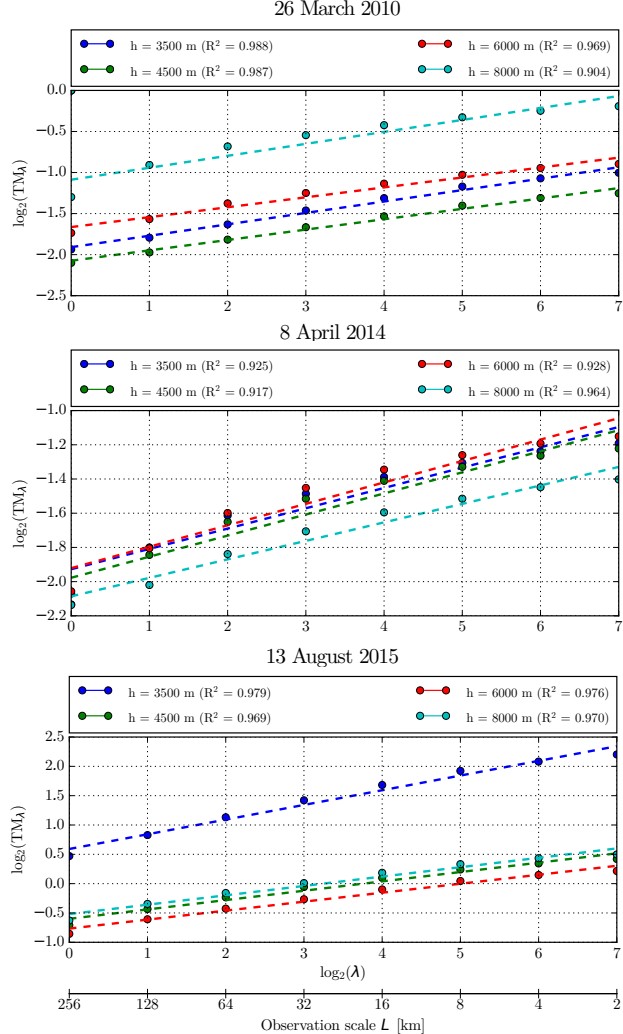

**Figure 7.** TM analysis of the ice water content (IWC) at different heights for $q = 1.5$. The dots indicate the measured moments and the line with same color corresponds to a linear best fit. The corresponding coefficient of determination $R^2$ is given in the legend.

considered in this comparative analysis due to the fact that it is convective and thus quite local in nature. In order to account for the displacement of the precipitation system from the East to the West during events 1 and 2, the simulation time was set six hours earlier for the flat region which is located more in the West.

The analysis of the power density spectra of the four water quantities (LWC, IWC, QV and TWC), reveals similar

5    features as in Section 4.1 for both regions. QV and TWC generally scale well at all altitudes and for both domains whereas IWC and LWC scale less and less with altitude ($\beta$ approaches zero). For the first event (26 March 2010) both regions show very similar scaling properties of IWC and LWC, with a typical progression from good scaling at




low altitude to poor scaling with a scaling break at around 8 km at higher altitudes to finally no scaling at all. For the second event (8 April 2014) a large difference can be observed in IWC with a scaling at much higher altitudes over the steep region. This might be due to the larger vertical extension of IWC over the Swiss Alps due to to the orographic lifting.

5    For the first event (26 March 2010), the flat regions seems to show a better scaling (larger $R^2$ in the TM analysis) than the steep region at least for IWC, LWC, and TWC. This can be seen in Figure 8 which shows the $R^2$ and $H$ values for the IWC. For the second event (4 April 2014), however the situation seems reversed. In terms of non-conservation parameter $H$, no general conclusions can be drawn, except that for both the flat and steep regions, the values of $H$ are large for IWC, TWC and QV. At least during the two studied events, the influence of the topography

10   on the non-conservativity of the COSMO water contents is not easy to characterize. The multifractal properties of the studied fields seem to be more influenced by the nature of the precipitation event than by the underlying topography. As such, the conclusions drawn in the previous section can be considered as relatively independent of the considered area.

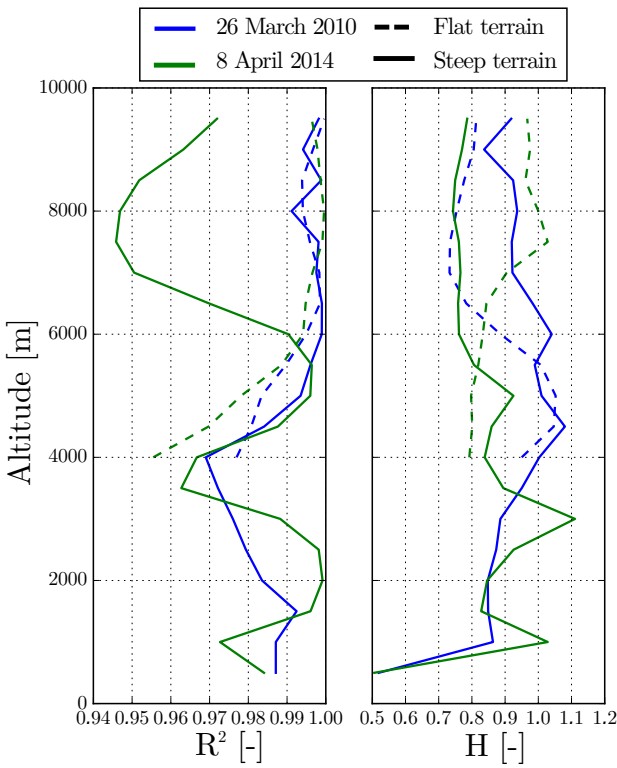

**Figure 8.** Evolution of the TM analysis coefficient of determination $R^2$ and the non-conservation parameter $H$ for the IWC as a function of height for the two considered events in flat and steep terrain.





To summarize, COSMO water contents generally show good scaling in time at all altitudes and temporal scales. In space however, IWC and LWC show a scaling break at around 8 km, with weak or even no scaling at larger scales. This break is particularly visible during the first and second events. The importance of this break increases with altitude, until no scaling can be observed any more ($\beta \approx 0$). Additionally, both IWC, QV and TWC are strongly

non-convervative at all heights and during all events (high $H$), and a simple correction (Equation 14) is generally not sufficient to make them conservative. A comparative study showed that the topography does not seem to play a major role on these conclusions.

## 4.3   Sample-based analysis

As pointed out in the previous chapter, the corrected fields of IWC, QV and TWC can not be considered as truly

conservative. It was decided however to still perform a DTM analysis in order to highlight global trends while keeping in mind that the retrieved values of $\alpha$ and $C_1$ might be overestimated, respectively underestimated. In the next section, this DTM analysis on water quantities will be extended with a multifractal study of precipitation intensities on the ground.

*Inter-comparison of multifractal parameters for the four water quantities*

Figure 9 focuses on the comparison of $\alpha$ for the four water quantities during the last (convective) event. It can be observed that $\alpha$ is generally larger for water vapour and total water content (which is dominated by water vapour) than for liquid water and solid water indicating a higher variability of these quantities. For the water vapour a layer of high $\alpha$ values (large variability) is present up to 4000 m altitude, which corresponds to the average height of the planetary boundary layer on that day. It can be observed that TWC and QV are characterized by a much larger

variability than IWC and LWC. For LWC this can be explained by the large number of zeros at high altitudes. At around 16:00 there is generally an increase in $\alpha$ for all water quantities which corresponds to the development of a large convective system over the study area.

*Comparison of one and two-moment schemes*

Figure 10 shows the multifractal parameters $\alpha$ and $C_1$ as well as the determination coefficient of the TM analysis

$R^2$ during the third (convective) event for the one and two-moment schemes. The third event is the one during which the discrepancies between the two microphysical schemes are the most obvious. It is interesting to notice that for the two-moment scheme a layer of large $\alpha$ and large $C_1$ is present at high altitudes (around 9000 m) from 17:00 to 21:00. This can be related to the fact that during this event the two-moment scheme tends to produce a larger number of small values of liquid water at high altitudes than the one-moment scheme. Generally in term of LWC,

the two-moment scheme seems to yield more variability for a longer time and at a larger vertical extension than the one-moment scheme. The quality of the scaling, characterized by the value of $R^2$ (last plot) also seems slightly better for the two-moment scheme, particularly at higher altitudes.

These observations can be extended to the other events where similar observations can be made, both in terms of larger $\alpha$ and $C_1$ in altitude and discrepancies in LWC average and number of zeros.





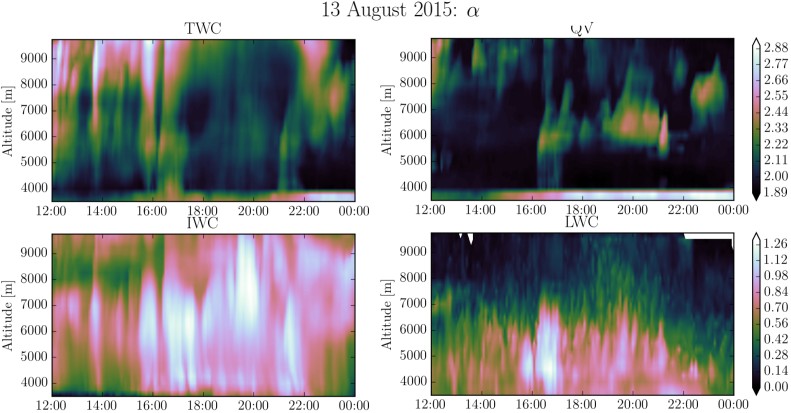

**Figure 9.** Evolution of $\alpha$ with altitude and time for the four water quantities during the convective event of the 13 August 2015. Note that the contour levels are different between LWC and IWC and TWC and QV.

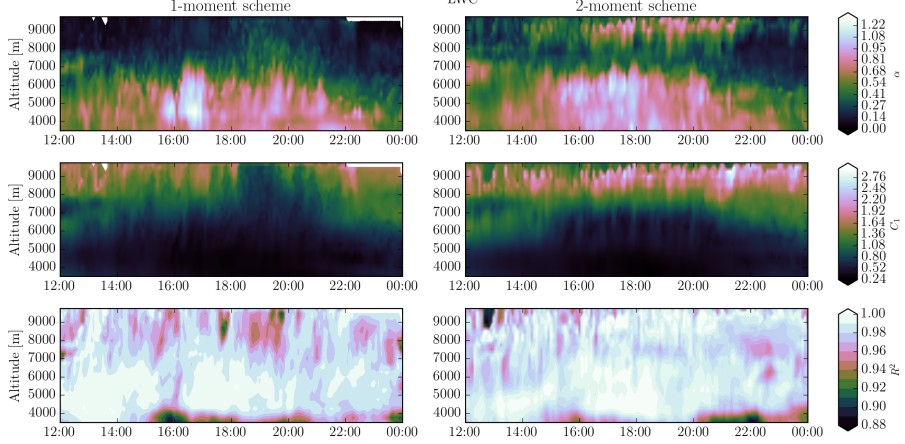

**Figure 10.** Evolution of $\alpha$, $C_1$ and $R^2$ with altitude and time for the liquid water content (LWC) during the convective event of the 13 August 2015

*Comparison of events*

Figure 11 shows the evolution of $\alpha$ with altitude and time for the liquid water content (LWC). For all events there is a generalized decrease of $\alpha$ over altitude, which goes along with an increase of $C_1$ due to the rarefaction of liquid water with altitude which causes an increase in the number of zeros. For the second event, this decrease of $\alpha$ with altitude is particularly rapid, which can be explained by the presence of a melting layer and the stability of the atmosphere during this event, where most liquid water is trapped below the melting layer (around 2400 m a.s.l). Finally, the last event is characterized by a higher vertical extension of large $\alpha$ values, which can be explained by




the presence of deep convection, which is especially strong from 16:00 to 17:00 (high CAPE values in Figure 4). The intensity of convection can be observed in the time series of $\alpha$ by higher values at high altitudes, which are absent for the second event (stratiform and stable) and very present for the last event. As such these time lines can be used as a diagnostic tool to quantify the amount of convection during a precipitation event.

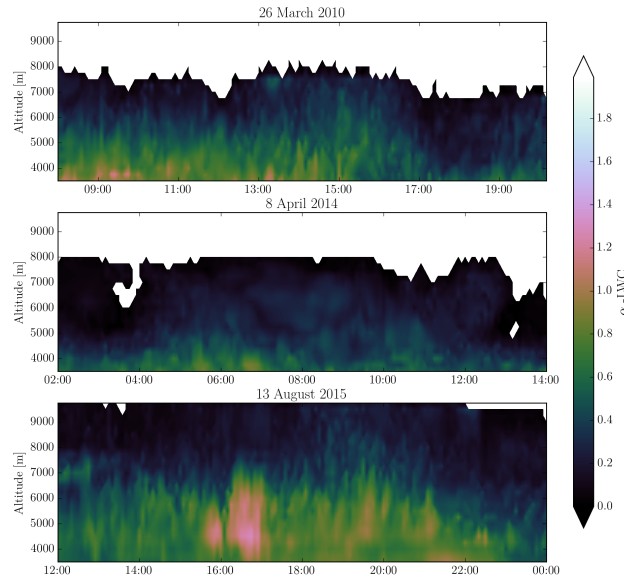

**Figure 11.** Evolution of $\alpha$ with altitude and time for the liquid water content (LWC) during all three studied events. The white pixels at high altitude indicate that no multifractal parameters could be computed, due to the field being all zero.

A comparison of the $\alpha$ and $C_1$ parameters estimated in time by considering an ensemble of 2D spatial fields (one for each time step) and estimated in space, by considering an ensemble of time series (one for each grid point) was also performed. It was observed that generally, the simple spatio-temporal model of Section 3.3 does not seem adequate to represent the multifractality of the water contents simulated by COSMO, which might be due in part to the fact that the fields are not truly conservative.

## 5   Multifractal analysis of precipitation intensities at the ground

### 5.1   Scaling analysis

A multifractal comparison of the precipitation fields simulated by COSMO in its one-moment and two-moments schemes with the QPE product from the Swiss radar composite was performed. As a first step, a spectral analysis was performed both in time (ensemble of one-dimensional time series of precipitation intensities) and space (ensemble
of two-dimensional maps of precipitation intensities).





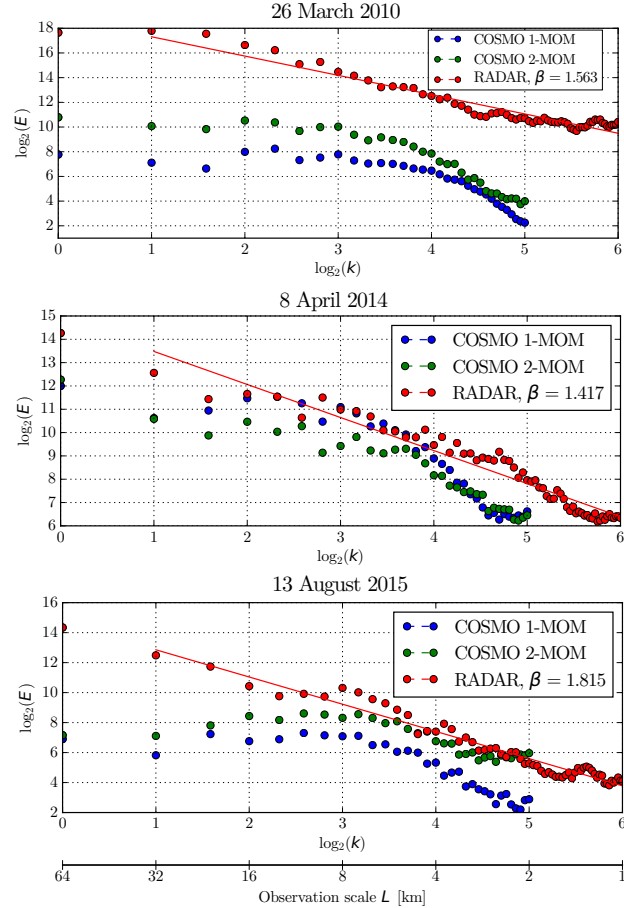

**Figure 12.** Spectral analysis in space of the QPE products during the three events. Bold lines are best-fit lines. The associated value of $\beta$ is given in the legend.

Figure 12 shows the spectral analysis in space for all events and data. A best-fit line is shown for the radar QPE from which the value of $\beta$ is computed (opposite of the slope). For the 26 March 2010, we observe a single scaling regime for the radar QPE, with a good scaling both at large and small scales, as the spread around the line is relatively small. For the model intensities, we observe strong discrepancies in spectral slope with the radar QPE at

5   smaller scales ($< 2 - 8$ km) which are not well represented. A possible explanation for this break in scaling properties of the model, is the fact that large scales are dominated by the dynamics of the model (primitive equations of the atmosphere) whereas smaller scales are dominated by the parametrizations of sub-grid phenomena (turbulence, convection). However, even at larger scales (8-64 km), the agreement between radar QPE and model simulations is still quite poor in terms of spectral slope. Obviously, for this rainfall event, COSMO is not able to recreate the

10   spatial structure of precipitation observed by the radar.





For the 8 April 2014, the scaling is similar between radar and model precipitation intensities, possibly indicating that for this stratiform rain event, parametrizations and dynamics match better. For the last event, we observe again a good scaling for the radar QPE and a much worse scaling on the model precipitation intensities, but in contrast with the first event, this time the larger scales ($> 8$ km) are not well represented. Indeed, inspection of the time series of precipitation shows that COSMO is not able to locate accurately the convective cells of precipitation and generally overestimates their extent. In terms of microphysical parametrizations, we observe that the spectral slopes of the one-moment scheme are generally closer to the ones obtained from the radar QPE, this is especially visible for the last (convective) event, where the two-moment scheme exhibits a weak scaling ($\beta$ close to zero). These observations agree with the scaling analysis of LWC and IWC, which consist mostly of precipitating hydrometeors (Section 4.1), where a scaling break was observed at large scale for the convective scale and at a smaller scale for the two other events. Note however that whereas the study of water contents was done at high altitudes (3500 m and more), the analysis of precipitation intensities is done at the ground level.

The spectral analysis in time (not displayed) shows generally similar results, but with larger values of $\beta$ and overall better scaling (less spread).

Analysis of the $H$ non-conservation parameters (Table 2) shows that in most cases, the fields simulated by the one-moment scheme are non-conservative in time and in several cases, even strongly non-conservative ($H > 0.5$). In space, there do not seem to be obvious trends in terms of comparison of H between model and data. Indeed the QPE $H$ is the smallest for the 26 March event, between the values found for the two model schemes for the April one, and the largest for the August one.

The radar QPE product seems to be generally more conservative than the COSMO simulations, with the exception of the last convective event. It is also worth noticing that the two-moment scheme is almost always more conservative than the one-moment scheme. In order to account for the fact that the fields are mostly non-conservative and to treat all fields in a consistent ways, all further analysis was performed on fluctuations of the original fields (Equation 14).

|  | 26 March 2010 | 8 April 2014 | 13 August 2015 |
|---|---|---|---|
| $H_{\text{space}}$ | 0.411/**0.432**/*0.752* | 0.342/**0.500**/*0.260* | 0.651/**0.612**/*0.332* |
| $H_{\text{time}}$ | -0.044/**0.615**/*0.262* | 0.232/**0.938**/*0.238* | 0.696/**0.818**/*0.265* |

**Table 2.** Values of the non-conservation parameter $H$ in time and space for all events, for the radar QPE, the COSMO **one-moment scheme** and the COSMO *two-moment scheme*.

Figure 13 shows the trace-moment (TM) analysis in time and space of the three events. For the two first events a scaling break can be observed at large scales for the COSMO intensities (64-128 km). These scales were excluded from the analysis, due to the limited number of points in this scale range. For the first event, in order to be consistent with the observations of the spectral analysis, the scale range 2-4 km, which does not scale well on COSMO simulations, when compared with radar observation was excluded from the analysis. For the last event, two scaling regimes are observed for the COSMO intensities, (2-16 and 16-128 km), which were studied separately. On the opposite, for

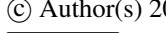


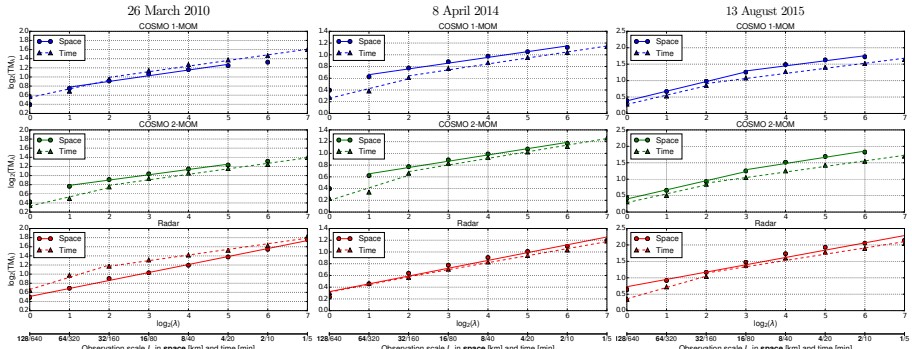

**Figure 13.** Scaling analysis of the QPE product during the three events. Bold lines are best-fit lines taking into account a possible scaling break.

the radar QPE no scaling break is observed in space. In time, a weak scaling break can be observed both for radar and COSMO intensities at a resolution of around 160 minutes. Hence results are discussed only for the time scales between 5-160 min (smaller scales).

## 5.2 Spatio-temporal analysis

Values of $\alpha$, $C_1$ and $\gamma_s$ obtained with an analysis in time and in space of the three events are given in Figure 14. For the first two events, all parameters are computed only on the smaller scales (up to 64 km in space and up to 160 minutes in time), in order to account for the observed scaling break. For the last events both scale ranges are considered.

  For the first event, both COSMO microphysical schemes give very similar multifractal parameters and the dis-
crepancy with the radar QPE is quite important. In space, it can be observed that $\alpha$ is slightly smaller in the COSMO simulations than on the radar QPE. It is clear as well that the simulated $C_1$ is too small compared with the radar observations. This tends to indicate that COSMO is underestimating the spatial intermittency. Generally,the observed discrepancies in $\alpha$ and $C_1$ tend to indicate that the spatial structure of the simulated fields is too smooth and lacks the variability observed by the radars. In time, the agreement is better for $C_1$ but COSMO has clearly
higher values of $\alpha$ indicating a larger temporal variability than the radar QPE. For this event, there is a noticeable discrepancy between the maximum singularity $\gamma_s$ in space obtained from the radar QPE (0.721) and the $\gamma_s$ obtained from the model (around 0.6 for both schemes). This indicates that during this event COSMO had a tendency to under-estimate extreme values, which might be caused by its difficulty to accurately simulate snowfall events, since COSMO does not consider partially melted snow (Frick and Wernli, 2012). Note that QPE in snow is very difficult
and it is likely that the radar QPE itself is already underestimating precipitation intensities ( Speirs et al. (2016), under review) which would make this difference in $\gamma_s$ even more noteworthy.





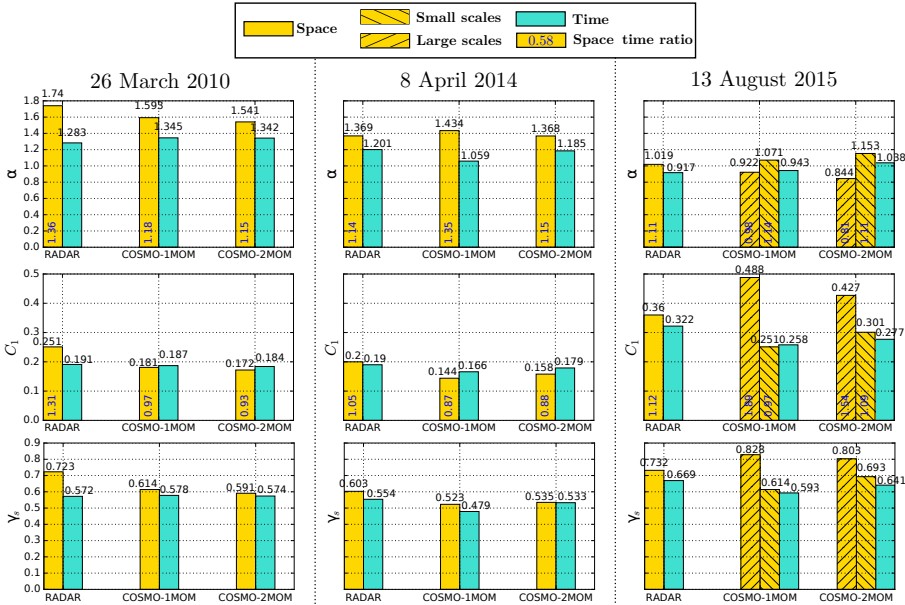

**Figure 14.** $\alpha$, $C_1$ and $\gamma_s$ parameter values obtained with an analysis in time and space for the three events on the fluctuations of the precipitation intensities. For the last event both the parameters at large and small spatial scales are displayed. The numbers in blue are the space/time ratios for $\alpha$ and $C_1$

For the stratiform rain event, the multifractal parameters of the COSMO simulations are in better agreement with the radar QPE. In time, the two-moment COSMO scheme gives values that are in relatively close agreement with the radar QPE and in this regard outperforms the one-moment scheme. COSMO simulations show generally smaller values of $\alpha$ and smaller values of $C_1$ than the radar QPE which is a trend that is observed for all events.

For the last convective event, two scaling regimes are considered in space, larger scales (16-128 km) and smaller scales (2-16 km). As already observed in the spectral analysis there is a better agreement between the radar observations and the simulations with the one-moment scheme at smaller spatial scales. In time however, the temporal intermittency of COSMO is smaller than for the radar QPE, which can be explained by the fact that COSMO generally overestimates the extent of the convective systems. Compared with the one-moment scheme and the radar

QPE, the two-moment scheme has a smaller $\alpha$ in space but a larger $\alpha$ in time, as well as a a smaller intermittency in time and space.

On the whole, the observations of the spatio-temporal analysis are consistent with the spectral and scaling analysis where (1) a strong discrepancy in scaling behaviour was observed between COSMO and the radar QPE at small scales for the first event,(2) a better scaling of the model precipitation intensities was observed for the second event,

(3) a discrepancy in scaling at large scales was observed between COSMO (especially for the two-moment scheme) and the radar QPE for the third event.




Overall, it can be observed that except for the first event where both schemes give similar values, the two-moment scheme is usually characterized by a larger $C_1$ than the one-moment scheme, both in time and space, whereas in terms of $\alpha$ there is no recurring trend. In terms of multifractal parameters $\alpha$ and $C_1$, there is generally a good agreement between radar observations and simulations on the range of scales were the model exhibits a good scaling behaviour, with none of the two microphysical schemes performing significantly better than the other. The two-moment scheme however is generally characterized by a slightly larger maximum singularity $\gamma_s$ indicating a better capacity to simulate extreme values. This is especially visible in the last convective event. In terms of space/time ratios, the observed ratios differ significantly from the theoretical model: the $\alpha$ space/time ratio is always larger and the $C_1$ space/time ratio always smaller than the theoretical values (1 and 1.44 respectively).

## 5.3 Timeseries of multifractal parameters

Figures 15, 16 and 17 show the timeseries of $\alpha$ and $C_1$ throughout the studied events for the COSMO and the radar QPE precipitation intensities, as well as some illustrative precipitation fields that will be discussed. For the third event only the parameters corresponding to the smaller scale range are used.

*26 March 2010*

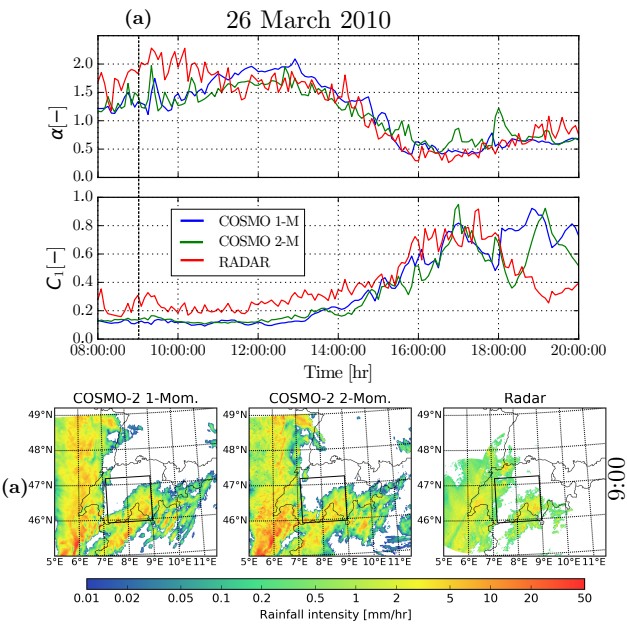

**Figure 15.** $\alpha$ and $C_1$ timeseries during event 1. The black box in the precipitation fields (bottom) corresponds to the square study area used to compute the multifractal parameters.

For the first event, before 11:00, which corresponds to a period of time during which the presence of graupel is important, both COSMO schemes perform similarly in terms of $\alpha$, which seems to be generally smaller than





on the radar observations. Starting from 11:00, until 15:00, the one-moment scheme has a larger $\alpha$ than both the radar QPE and the two-moment scheme. In terms of $C_1$, before 18:00, we observe a consistent positive bias between observations and simulations, independently of the microphysical scheme that is being used. As observed previously in the spectral analysis, this bias is caused by the fact that the model is unable to take into account small-scale

structures of the precipitation system. An example can be seen in panel ($a$) in Figure15, on the east of the study domain, where the radar detects many small-scale precipitation cells that are not accurately resolved by the model. Finally, starting from 13:00, there is an increase in the mean intermittency over time as the system progressively leaves the observation domain.

*8 April 2014*

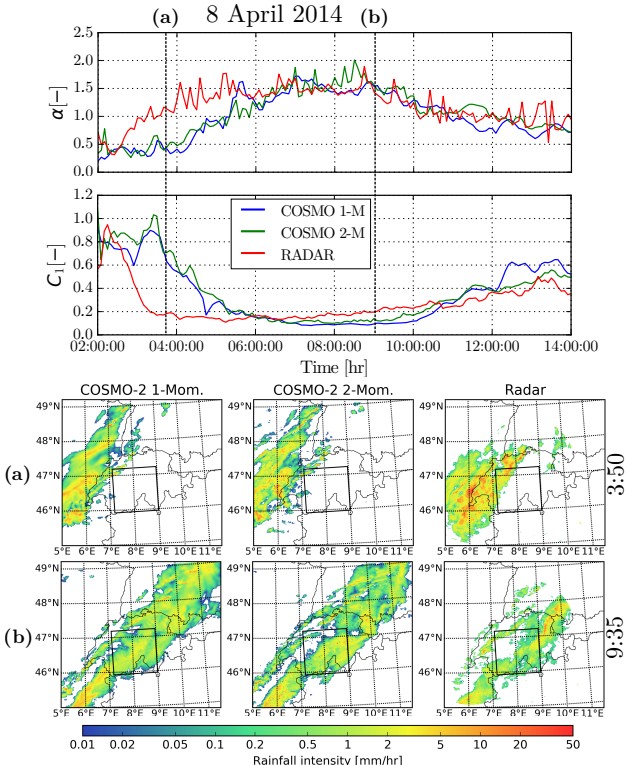

**Figure 16.** $\alpha$ and $C_1$ timeseries during event 2.

During most of this event, both $\alpha$ and $C_1$ agree well between the one-moment and the two-moment schemes. The simulated fields have generally a lower $\alpha$ and a higher $C_1$ than the radar observations, especially during the first half of the event when the precipitation system is not fully developed over the domain. This difference can be explained by a shift to the west of the simulated field compared with the observations, which is particularly visible between 3:00 and 5:00 and dissipates later on. The center of the precipitation system where the precipitation intensity is

larger being absent, this leads to a lower $\alpha$ in the simulation (for example in panel ($a$) in Figure 16) and a larger $C_1$





caused by a larger number of pixels without precipitation. This temporal shift gets attenuated during the simulation and toward the end of the event, simulation and observation have similar multifractal parameters, with however a better agreement of the two-moment scheme in terms of $\alpha$. This can be verified in panel $(b)$ of Figure 16 where the one-moment scheme simulation appears too smooth when compared with the observation.

5    *13 August 2015*

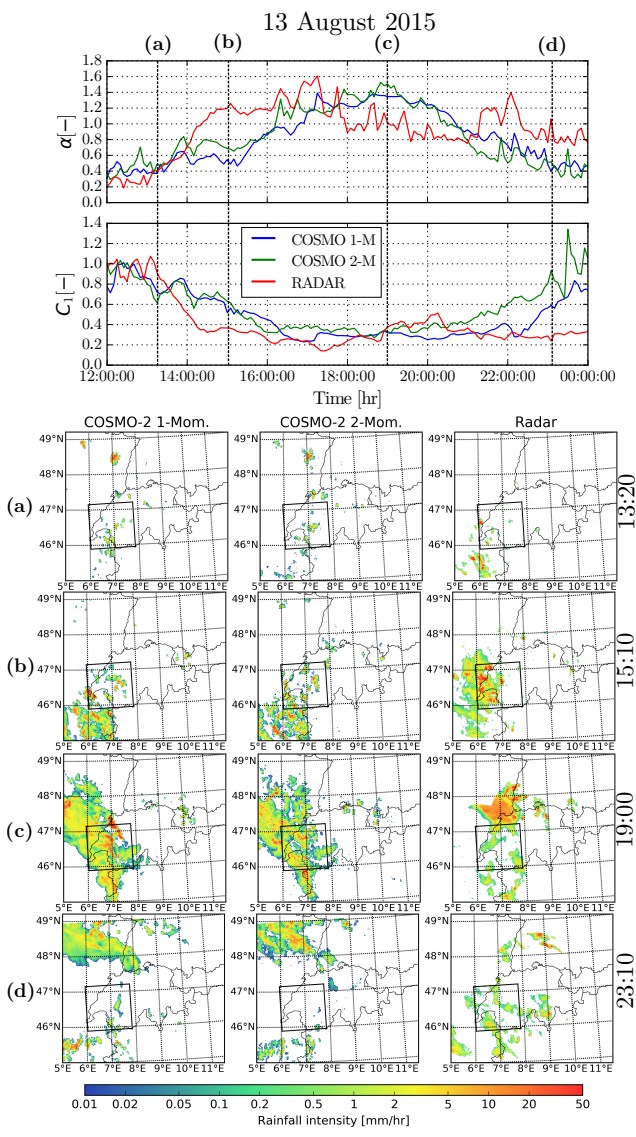

**Figure 17.** $\alpha$ and $C_1$ timeseries during event 3.



For the convective event, four different phases can be identified. In the first short phase (12:00-14:00), observations and simulations agree relatively well in $\alpha$ and $C_1$. This period corresponds to the initial stages of the event when only a few isolated cells are present (panel ($a$) in Figure 17).

In the second phase (14:00 - 17:00), a large convective system is crossing the domain on the radar observations, which causes a strong increase in $\alpha$ and a decrease in $C_1$. This convective system is however located more in the south on the simulation and enters the domain only at around 15:30 (panel ($b$) in Figure 17).

During the third phase (17:00-21:00), the large convective system is visible on the simulated field, whereas on the observed radar fields, the most intense convective cells are already out of the domain. This causes a larger $\alpha$ on the simulations than on the observations (panel ($c$) in Figure 17). The effect of such shifts on the multifractal analysis hints at the possibility of a further analysis based not on a fixed study domain but on a study domain following the precipitation system, in a way similar to Nykanen and Harris (2003). Finally in the last phase (21:00-24:00), a new convective system is visible on the observed field but is more or less absent on the simulated fields. This causes a discrepancy, the simulated fields having a smaller $\alpha$ and a larger $C_1$ than the observations (panel ($d$) in Figure 17). As stated previously, the spatial and temporal shift of the convective system simulated by COSMO with respect to the radar observations is the main cause in the bad scaling observed at larger scales.

This succession of phases is also clearly visible in the timeserie of wet area fraction (Figure 18)

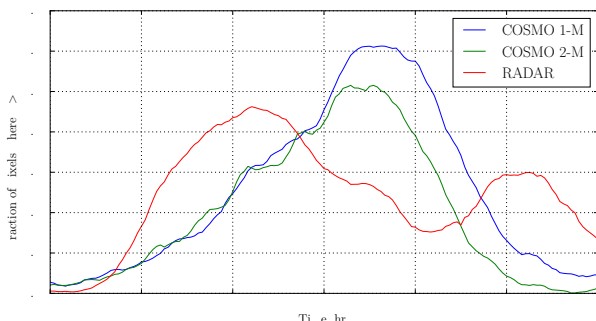

**Figure 18.** Fraction of wet area during the event of the 13 August 2015

## 6    Conclusions

In this work we performed a spatial and temporal analysis of the variability of precipitating and non-precipitating water contents simulated by the COSMO NWP model, in the context of the universal multifractal (UM) framework which allows to represent the variability across scales with a limited number of parameters. The analysis focused on three different events, one cold front associated with heavy snowfall, one stationary front associated with stratiform rain and a stable atmosphere and one summer convection event with heavy rain. All events were simulated at a 2





km resolution with both the standard operational one-moment microphysical parameterization of COSMO and a more advanced two-moment microphysical scheme.

The multifractal analysis of the water contents in liquid, solid and gas phases reveals that these quantities are indeed multifractal but that only the water in liquid phase (LWC) is a conservative quantity. The LWC displays a

scaling break around 8 km, which might be a consequence of the model parametrization of sub-grid processes. Other quantities generally show a single scaling regime at all altitudes. Unfortunately the simple differentiation method (Equation 14) was not sufficient to yield conservative fields and as such the DTM analysis could only be used to indicate possible trends: the two-moment scheme seems characterized by a higher variability than the one-moment scheme (1), the time series of the multifractal parameters and their vertical extent are in good agreement with the

atmospheric instability represented by the CAPE (convective potential energy) (2). Finally, the influence of the topography on the non-conservativity parameter $H$ and DTM parameters was found to be difficult to characterize in a simple way, with no recurring trend between events, indicating that the variability of the multifractal parameters is dominated by the nature of the precipitation event and not the underlying topography.

The second part of this work focused on the multifractal comparison of precipitation intensities at the ground

simulated by COSMO with the Swiss radar composite data. Whereas the radar data shows one single scaling regime over the studied spatial scale ranges (1-128 km), the COSMO simulations display scaling breaks for the first and the last event. It can be observed that during the snowstorm event COSMO is unable to properly reproduce radar observations at small scales, which might be caused by the intrinsic difficulty of simulating solid precipitation. During the last convective event, the opposite can be observed, and COSMO is struggling to reproduce the larger scales,

due to its difficulty to locate properly the convective system in time and space during this event. In the temporal scales, a scaling break is observed both for the radar and COSMO simulations at around 3 hours.

Comparisons of the one-moment and two-moment COSMO microphysical parametrizations show that the fields simulated by the two-moments scheme tend to display a larger intermittency and variability than the one-moment scheme. However, this does generally not translate into a better agreement of multifractal parameters with the radar

composite, except during the stratiform event where the two-moment scheme performs slightly better.

Ultimately, the multifractal framework can be used to identify the scale ranges in which the model is able to simulate realistic fields of water contents and as such this technique can be used as a diagnostic tool for model evaluation.

*Acknowledgements.* The authors would like to thank MeteoSwiss for the access to the Swiss operational radar composite as

well as the initial and boundary conditions used in the COSMO simulations.

The authors thank the Partenariat Hubert Curien – Germaine de Staël (Projet 32709UK) for financial support that made this collaboration possible.



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
