# Peer review of "Multifractal evaluation of simulated precipitation intensities from the COSMO NWP model"

_Atmospheric Chemistry and Physics, 2017_

## Referee Comment (RC1) · Anonymous Referee #1 · 14 Mar 2017

**General comments**

Wolfensberger et al. interpret precipitation pattern as universal multifractals and explore the feasibility of this approach with regards to (i) investigating the sensitivity of precipitation pattern to orography and the choice of the cloud microphysics scheme (ii) evaluating a NWP model with observations. Multifractal methods have never been presented in ACP so that an application of this technique within the scope of atmospheric chemistry and physics is very interesting. The analysis of Wolfensberger et al. is somewhat unsatisfactory, however. It remains in large parts descriptive and only touches on interpreting the results of the multifractal analysis in relation to the underlying dynamics and physics. In particular, it does not become completely clear, what practical insights

can be gained from the multifractal analysis as compared to the simple scaling analysis. By elaborating on these issues, the manuscript could be strongly improved and make the potential of multifractal analyses accessible to a broad readership of ACP.

**Specific comments**

- The analysis of liquid water contents (Section 4) seems to be strongly hampered by the non-conservativity of the fields. Given also the weak conclusions on the sensitivity to orography ("the dynamics of the weather event are more important than orography") and microphysics scheme ("the more complex scheme results in more variability"), it might be worth a thought if this section really strengthens the analysis or if the paper could be reduced to the surface analysis in Section 5.
- Given that many readers of ACP might not be familiar with multifractals, it might be helpful to expand Section 3 with an example that shows how the appearance of a field changes for changing values of  $\alpha$ ,  $C_1$  and  $\gamma_s$ .
- What can be learned from the (non)-conservativity of a field?
- The locations of the scaling breaks differ between the scaling and the multifractal analysis. What are the corresponding interpretations and which scale should, e.g. a model developer take into account when trying to identify the responsible model process?
- Neither the model nor the radar data are in agreement with the simple space-time scaling model. This result should be discussed, especially in view of the correspondence between CAPE and multifractal parameters. This correspondence indicates a close relationship between precipitation pattern and dynamics, similar to the assumption underlying the simple space-time scaling model. In addition,

the agreement found by Gires et al. (2011) for Meso-NH and corresponding radar data should be addressed.

• I could not quite follow the interpretation of Fig 12 and Table 2 (see below). For Fig 14, I wonder how relevant (although presumably significant) observed differences are (see below).

**Technical corrections**

- p7, L16 and 18: Should this read Zone 2 and Zone 3 instead of Zone 1 and 2?
- p9, L11: Isn't this an upper threshold that the values of  $\epsilon_{\lambda}$  fall below rather than to exceed it?
- p9, L13:  $c(\gamma)$  instead of  $c(\lambda)$
- p9, Eq 5:  $K_c$  instead of K
- p6, Eq 6: What is *D*?
- p12, Eq 14: no italics for subscript "time"
- p13, Fig 5: indicate that the different colors correspond to different levels
- p14, L5-L8: refer to zones in Fig 2
- p16, L7: April 8 instead of 4
- p16, Fig 8: Is the legend for flat and steep terrain flipped here?
- p17, L29: terms instead of term
- p19, L7: refer to Eq. 13 or 14, respectively

**C3**

- p20, L2: To me, there seems to be a scaling break in the radar data at 4km
- p21, L6: Doesn't the spectrum show an under-representation of large features in the model as compared to the radar?
- p21, L8: Both, the one and two moment scheme have  $\beta \approx 0$  for the last event.
- p21, L16: several cases = all except for one
- p21, L18: In space, QPE H is smallest for April 8 with a value of 0.342, not for March 26
- p21, L23: Equation 12, not 14; I assume, this time the resulting fields are conservative? This should be mentioned.
- p22, L7: event instead of events
- p22, L14: How meaningful is a "clear difference" of 1.34 or 1.35, respectively, compared to 1.28 for practical purposes? Asked differently, what is the accuracy of these values?
- p28, L7: Equation 12 instead of 14

---

## Author Comment (AC1) · 30 Jul 2017

article [a4paper, total=6in, 8in]geometry xcolor

[Figure]

**Answer to reviewers**

Daniel Wolfensberger, Auguste Gires, Ioulia Tchiguirinskaia, Daniel Schertzer and Alexis Berne

July 30, 2017

The new version of the paper is provided as a supplement to this comment. The new parts are shown in blue, the discarded parts are not shown (to make it easier to read).

**1 Review 1**

**1.1 General comments**

Wolfensberger et al. interpret precipitation pattern as universal multifractals and explore the feasibility of this approach with regards to (i) investigating the sensitivity of precipitation pattern to orography and the choice of the cloud microphysics scheme (ii) evaluating a NWP model with observations. Multifractal methods have never been presented in ACP so that an application of this technique within the scope of atmospheric chemistry and physics is very interesting. The analysis of Wolfensberger et al. is somewhat unsatisfactory, however. It remains in large parts descriptive and only touches on

interpreting the results of the multifractal analysis in relation to the underlying dynamics and physics. In particular, it does not become completely clear, what practical insights can be gained from the multifractal analysis as compared to the simple scaling analysis. By elaborating on these issues, the manuscript could be strongly improved and make the potential of multifractal analyses accessible to a broad readership of ACP.

**1.2 Specific comments**

1. *The analysis of liquid water contents (Section 4) seems to be strongly hampered by the non-conservativity of the fields. Given also the weak conclusions on the sensitivity to orography ("the dynamics of the weather event are more important than orography") and microphysics scheme ("the more complex scheme results in more variability"), it might be worth a thought if this section really strengthens the analysis or if the paper could be reduced to the surface analysis in Section 5.*

   We have removed this part of the paper and replaced it with a climatological study of simulated precipitation intensities using the universal multifractal framework, we hope that this will illustrate in a more intuitive way the relation between multifractals and atmospheric variables.

2. *Given that many readers of ACP might not be familiar with multifractals, it might be helpful to expand Section 3 with an example that shows how the appearance of a field changes for changing values of $\alpha$, $C_1$ and $\gamma_s$.*

   We have added two plots (Figures A.1 and A.2 in the appendix) which illustrate the effect of changing $\alpha$ and $C_1$ and $H$ on the appearance of a two-dimensional multifractal fields.

3. *What can be learned from the (non)-conservativity of a field?*

A value of $H$ larger than 0 indicates that the field is smoother than the observed field from a direct multifractal cascade process and a value of $H$ smaller than 0 indicates that the field is too discontinuous. When comparing $H$ between COSMO and the radar QPE, one frequently observes that $H$ is larger on the simulations, indicating that the spatial structure of the simulated fields is likely to be too smooth. We have tried to emphasize this in the text, when talking about the value of $H$.

4. *The locations of the scaling breaks differ between the scaling and the multifractal analysis. What are the corresponding interpretations and which scale should, e.g. a model developer take into account when trying to identify the responsible model process?*

   In the QPE part, the scaling breaks agree quite well between the spectral analysis (Figure 7) and the TM analysis (Figure 8). Part of the discrepancy you refer to was coming from the fact that the x-axis of Figure 7 was wrong. This has been fixed now (Observation scales goes from 2 to 128 instead of 1 to 64).

5. *Neither the model nor the radar data are in agreement with the simple space-time scaling model. This result should be discussed, especially in view of the correspondence between CAPE and multifractal parameters. This correspondence indicates a close relationship between precipitation pattern and dynamics, similar to the assumption underlying the simple space-time scaling model. In addition, the agreement found by Gires et al. (2011) for Meso-NH and corresponding radar data should be addressed.*

   Concerning the correspondence between CAPE and MF parameters, since this part has been removed from the paper there is no need to address the issue. In terms of agreement found with Gires et al., a new paragraph has been added in the paper at the end of Section 5.3:

   Gires et al. (2011) found different breaks for a Cevenol event (strong precipitation

events occuring in Fall in the South of France), i.e. roughly 16 km in space and 1h in time, and a better agreement with a simple space time model but only for large scales which are not the primary focus of this study. These differences could be associated with the fact that the topography of the area analysed in this paper is more pronounced than in **?**. It should also be noted that the values of UM parameters $\alpha$ and $C_1$ on the relevant range of scales exhibit a better agreement between observations and model simulations in this paper.

6. *I could not quite follow the interpretation of Fig 12 and Table 2 (see below). For Fig 14, I wonder how relevant (although presumably significant) observed differences are (see below).*

Since the data that is used is not instrumental (i.e. model data so not affected by noise), the only source of uncertainty comes form the numerical estimation method of multifractal parameters. The DTM method is generally quite robust, and if there is good scaling $R^2 > 0.95$, the uncertainty of the multifractal parameters would be negligible. In our case, since we used only the scale ranges with sufficiently good scaling we get values of $R^2$ that are usually very close to 1, with the smallest being 0.92. So though we cannot precisely state the uncertainty associated with the MF parameters, we think that are quite trustful. We have added the following sentence at the end of section 5.1

Note that in the considered range of scales the quality of scaling measured by the $R^2$ parameter is quite good (average $R^2$ in space = 0.963 ± 0.024, in time 0.956 ± 0.017. This implies that the uncertainty associated with the $\alpha$ and $C_1$ parameters retrieved with the DTM method is small.

**1.3 Technical corrections**

1. *p7, L16 and 18: Should this read Zone 2 and Zone 3 instead of Zone 1 and 2?*

This has been corrected and adjusted to agree with the modifications of the first part of the paper

2. *p9, L11: Isn't this an upper threshold that the values of $\epsilon_\lambda$ fall below rather than to exceed it?*

   Indeed, thank you for pointing this out. The sentence has been corrected

3. *p9, L13: c($\gamma$) instead of c($\lambda$)*

   This was corrected, thanks

4. *p9, Eq 5: $K_c$ instead of $K$*

   Yes, we added the subscript

5. *Eq 6: What is $D$?*

   Thank you for pointing this out, $D$ is the dimension of the field (1 for a timeserie, 2 for a spatial field). We have added this sentence in the text.

   $D$ is the dimension of the field (1 for a timeserie, 2 for a spatial field)

6. *p12, Eq 14: no italics for subscript "time"*

   Fixed

   Points 7 to 12 correspond to parts that has been removed so these points are not relevant anymore

13. *p20, L2: To me, there seems to be a scaling break in the radar data at 4km*

    There is indeed a scaling break at around 8 km (and not 4 km since as stated before there was an error in the x-axis labels) for the radar QPE for the second event only. We added this info. For the other events we do not see a scaling break with a comparable intensity to the scaling breaks observed for the COSMO precipitation.

Both radar and simulations show a weak scaling break at around 8 km.

14. *p21, L6: Doesn't the spectrum show an under-representation of large features in the model as compared to the radar?*

Yes we have added the following explanations in the text:

This is especially visible for the last (convective) event, where the COSMO simulations show weak scaling ($\beta$ close to zero). This implies that the simulated rainfall intensities are dominated by small-scale features, while large scale features are underestimated. Note also that for large scale features, the power density function of COSMO simulations correspond to white noise, indicating that the COSMO model has a shorter decorrelation range than the radar data.

15. *p21, L8: Both, the one and two moment scheme have $\approx$ 0 for the last event.*

Yes this quite true, we have changed the sentence accordingly, see last point.

16. *p21, L18: In space, QPE H is smallest for April 8 with a value of 0.342, not for March 26*

Yes indeed, the sentence was ambiguous. We only wanted to say that when compared with COSMO H values, then for the first event, the radar H is the smallest of the three. We have made this more clear in the text

Taking the radar as reference, one sees that the convective event is characterized by the largest values of $H$ followed by the snowfall event and the stratiform event.

17. *p21, L23: Equation 12, not 14; I assume, this time the resulting fields are conservative? This should be mentioned.*

We fixed the reference to the equation. We have also added the following sentence at the end of the paragraph to make it more clear:

Note that while this does not result in perfectly conservative fields, it still makes them more conservative since all values of $|H|$ are smaller than 0.5 after taking the fluctuations.

18. *p22, L7: event instead of events*

    Fixed

19. *p22, L14: How meaningful is a clear difference of 1.34 or 1.35, respectively, compared to 1.28 for practical purposes? Asked differently, what is the accuracy of these values?*

    See last point of the *Specific comments* section.

20. *p28, L7: Equation 12 instead of 14*

    See Point 18

**2  Review 2**

**2.1  Major comments**

1. *As noted above, universal multifractals is something not seen in most ACP publications. Thus,would highly recommend a table outlining the variables of interest and how they are tied to the analysis. In the text, it would be very beneficial to include a subsection that outlines what each variable means in terms of increases and decreases and relate this back to the physical context of the systems of interest.*

   We have added a table (Table 3) that gives an overview of all relevant parameters in the universal multifractal framework as well an interpretation of their effects on precipitation. In the appendix we have also added some illustrations of how this multifractal parameters affect the structure of a spatial field. We have also thoroughly changed the first part of the paper in order to better explain the link between the MF parameters and the meteorological and geographic variables.

2. *Again, as noted above, much of the analysis is cursory at best, leaving the reader scratching his/her head for an explanation. For example, Lines 15-19 on Page 21 and Lines 10-11 on Page 22. There are many more. I kept asking myself why? I realize that the paper is already a bit long and so adding more analysis will make it cumbersome to read. However, perhaps it is worth excluding the effects of topography and microphysics and focus more on the model-observation comparison? Leave the effects of topography and microphysics for a subsequent publication once the groundwork is published, especially since the results for these aspects seem case-dependent.*

   We have decided to keep the microphysical aspect as we thought that it would be interesting to see how the same model could produce quite different spatial structures depending on its configuration and since the paper is much shorter

now than it was before. The effect of topography is considered in the first part of the paper which is new but it replaces the whole section about the water contents (first part) which has been discarded as it was not very conclusive. Some parts have been shortened as well such as the very last part (study of the timelines of multifractals) which was quite redundant between events.

3. *A follow on to the previous point is that there are many instances where I felt as if there should have been a figure to reference and yet nothing was referenced. For example, the paragraphs beginning on Line 4 of Page 15, Line 33 of Page 17, and Line 1 on Page 21. There are other instances as well where it was not clear if a figure in the text was intended to be referenced or if analysis was conducted and not presented in the text. Along the same lines, some of the figures are very difficult to read due to the small font. Moreover, while I realize that the observations are difficult to retrieve at low elevations, the model can and does simulate such levels. I was a bit puzzled as to why these levels were removed from the initial analysis. Perhaps they should be included why just examining the model and then removed for comparisons with observations.*

The issue with missing model levels was coming from the fact that the multifractal framework does not handle missing data which naturally occur when interpolating model data at fixed altitudes. In order to addres this limitation, we have now focused the first part of the paper on the precipitation at the ground level as simulated by the model, so the lower troposhere is not ignored anymore.

4. *Lastly, I could not understand Table 2; some background and explanation is clearly warranted.*

We have provided a better description of the contents of Table 2 (Now Table 4)

Table 4 displays the non-conservation parameters H evaluated for timeseries of precipitation intensities (analysis in time) and for spatial fields of precipitation intensities (analysis in space), for both the radar QPE (in regular font), the COSMO

one-moment scheme (in bold) and the COSMO two-moments scheme (in italic) and for all events

**2.2   Minor comments**

1.  *The organization is quite nice; however, there are several grammar errors (especially punctuation), and there are issues with figure and equation referencing. Moreover, units are inconsistent (e.g., g versus mg).*

    We have checked all the references and have fixed the problematic ones, it should be fine from this point of view now. In terms of units they should be consistent now. The first part of the paper has been changed and we only deal with precipitation now, either in mm (for accumulated quantities) or mm $\cdot$hr$^{-1}$ (for precipitation intensities). Note however that multifractals parameters are not sensitive to units, so even using inconsistent units should not impact the results.

2.  *Consider not capitalizing words like east and west.*

    We have put all cardinal directions in lowercase

3.  *I would recommend including references throughout section 2.1 for all assumptions that go into the model.*

    Unfortunately most of COSMO's parameterizations have not been published in peer-reviewed jour- nals. We have included all references to peer-reviewed work we could find (this includes a few new references, e.g. Rutledge et al 1983 and Lin et al. 1983

4.  *Consider using section instead of chapter*

    The part where the word "chapter" was used has been removed from the paper so this is not an issue anymore.

5. *In equation 1, the variables do not match with the subsequent descriptions*

   Yes indeed, thanks for pointing this out, we changed the $\lambda$ in the equation to a $\Lambda$ in order to avoid confusion with the resolution in the multifractal framework. We have now fixed the description to use $\Lambda$ as well.

6. *Line 29 on Page 3: Should this be number concentration?*

   Yes indeed this is a better choice. We have changed "concentration" to "number concentration" and "mass fraction" to "mass concentration" to be more explicit and consistant.

7. *Line 30 on Page 3: Should this be mass mixing rations?*

   *See Point 6*

8. *Table 1: Consider writing out number*

   This has been fixed accordingly

9. *Line 8 on Page 5: What is meant by size being a power of two?*

   This paragraph is not in the paper anymore

10. *Line 4 on Page 7: Correct the definition of PPI.*

    We have changed this to "plane position indicator (PPI) "

11. *Lines 6-7 on Page 17: Reword to improve clarity.*

    As this part is not in the paper anymore, this issue is not relevant anymore

12. *Line 2 on Page 20: What is meant by opposite of the slope.*

    We have tried to make this more explicit by writing:

[revised manuscript text omitted]

---

## Author Response (AR2)

**Answer to reviewers**

Daniel Wolfensberger, Auguste Gires, Ioulia Tchiguirinskaia, Daniel Schertzer and Alexis Berne

October 20, 2017

**1 Associate Editor**

We thank the associate editor for his helpful comments. We have performed the request changes.

1. *There are quite a few technical, typographical errors that have to be taken care of. There is a general issue of readability because of English language errors. Please have a native speaker correct and run a spell checker.*

   The paper has been carefully reread and numerous errors have been corrected. The article has also been corrected by a native speaker (Tim Raupach), and we are quite confident that it has gained significantly in readibility.

2. *Caption to Table 1 is wrong*

   This has been fixed. The caption now reads:

   List and description of all synoptical and topographical descriptors used in the multifractal characterization of the climatology of precipitation intensities.

3. *I wonder whether how you calculate your rain rates in COSMO using the exponential functions and fixed slope parameter (single moment code) is consistent with the radar Z-R relationship.*

   Thank you for this very relevant remark. Indeed, we didn't think enough about this issue. We have now added a whole paragraph dealing with this issue at the beginning of Section 4.

   The power-law $Z - R$ relationship by (Marshall and Palmer, 1948), which is used by MeteoSwiss to derive the QPE, does not correspond with the $Z - R$ relationships derived from the COSMO microphysical parameterizations. This explains part of the discrepancy between radar QPE and simulated precipitation intensities, but in general should not impact the validity of the multifractal comparison. Indeed, if the $Z - R$ relationship derived from the COSMO parameterizations can be approximated by a power-law, then the correction needed to account for discrepancies in $Z - R$ relationships is itself a power-law: $R_{\text{corr}} = aR^b$, where $R_{\text{corr}}$ is the precipitation intensity one would obtain by first converting COSMO precipition intensities to reflectivities and then back to precipitation intensities using the radar QPE $Z - R$ relationship. It can be shown (Tessier et al., 1993) that in the context of universal multifractals, the corrected field will have the same value of $\alpha$ and the same scaling properties as the original field, while $C_1$ will be multiplied by $b^\alpha$. Moreover, for the one-moment scheme, it was observed that while the intercept parameter $a$ changes significantly, the exponent parameter $b$ is almost the same: $R_{\text{corr}} = 0.68R^{0.98}$. As the exponent 0.98 is close to unity, this implies an almost direct proportionality, and as such even $C_1$ should barely be affected. Note that this power-law was

derived by using the T-matrix method (Mishchenko, 1996) to compute radar cross-sections at C-band.

For the two-moment scheme, things are more complicated as no one-to-one relationship exists between rainrate and reflectivity. However, a rough estimation of the error on $C_1$ was derived by considering a representative set of rainy time steps from all events. The estimated values of $b$ for the two-moment scheme varied between 0.81 and 1.23, which would imply a maximum relative error on $C_1$ of 51% on spatial fields and 23% on time series.

Overall, correcting precipitation fields for discrepancies in $Z - R$ relation is a difficult task, especially in the solid phase and for the two-moment scheme, where deriving a $Z - R$ relation from the model parameterization is difficult. However, in the multifractal context, only $C_1$ should be affected and only with significant solid precipitation or when using the two-moments scheme. This should be kept in mind when interpretating $C_1$ values.

4. *Please pay attention to your section numbering, which seems to have failed in the track change version.*

   Unfortunately this issue appeared when preparing the reviewed paper version (with blue fonts) using a macro. We apologize for not seing this when submitting the reviewed paper. This has of course been fixed now.

**2 Anonymous Referee #1**

We thank the anonymous referee #2 for his helpful comments and review. We have performed the required changes whenever it was possible.

**2.1 Specific comments**

1. *- In my opinion, the climatological analysis could be greatly simplified and streamlined by directly using the Köppen classification as descriptor instead of the meteorological and geographical parameters. As shown in Figs. 3 and B3, the meteorological and geographical parameters used in the climatological analysis are not independent: Potential vorticity is highly correlated to altitude and temperature to latitude. It would therefore be enough to use either the meteorological or the geographical set of parameters - or, as suggested, the Köppen classification, which both parameters sets are also related to.*

   It is indeed possible to retrieve a similar classification by aggregating some of Köppen classes, however the choice of the aggregation could be perceived as somewhat arbitrary. Additionally, the Köppen classification is available only over land areas. We have thus decided to stay with the original classification but have added a paragraph in the appendix (C3), where both classifications are compared, and which shows that the obtained results are quite consistant.

2. *- The above suggestions would exclude the characteristics of the precipitation distribution (total amount, variance, wet fraction) as descriptors. Especially the wet fraction is currently used to explain the relationship between climatological classification as determined by latitude and altitude to the multifractal parameters. I am not fully convinced by the corresponding argument because the precipitation characteristics seem to be already "half-way" to the multifractal parameters. I would instead try to explain the multifractal parameters based on the typical precipitation types that correspond to the Köppen classification (e.g. frontal). Also the case studies from the second part might be useful here.*

Considering the small COSMO-2 area, most areas will be typically affected by frontal precipitation.

3. - *The TM coefficient is not included on p. 16, l 3 because it is not a real parameter. Why is it shown and discussed in Figs. 3, 5 and 6? The paper does not seem consistent in terms of the multifractal parameters investigated: The fractal dimension D_l is only used in the first part (without introduction) and Fig. 10 does not discuss $\gamma_s$. Is there a specific motivation for these choices?*

Our claim that $R^2$ is not a proper characteristic of an area was probably exaggerated. Indeed the quality of the scaling of precipitation within an area and how much the precipitation in this area is multifractal can be considered as a relevant characteristic. We have thus decided to also include this parameter in the statistical comparison. It appears that the $R^2$ is signicantly different between all clusters, so there is no explicit mention of it in the text, since only the non-significant parameters are mentioned.

Thank you for pointing out that we forgot to include $D_f$ (fractal dimension) in our theoretical explanations (Section 3) and in our second part of the results section. We have restructured the theoretical explanation to start with the definition of $D_f$ and then generalize the theory to multifractals, which should be easier to folow.

Let $\epsilon$ be a normalized (divided by its mean) conservative field, which can be one or two dimensional (time serie or spatial map). The fractal dimensions $D_f$ of a field indicates how the binary field (where all values larger than a given threshold are set to 1) scales with the resolution. The resolution is defined by the ratio between outer scale $L$ and observation scale $l$ ($\lambda = L/l$).

$$N_\lambda = \lambda^{-D_f} \tag{1}$$

where $N_\lambda$ is the number of positive samples (rainy pixels for example) at a given resolution, which can be obtained with the help of box counting.

It is possible to interpret this result in a probabilistic way. Indeed, let's consider a line or cube of size $l$. Pr is the probability that it intersects the field. This probability scales with the resolution:

$$\mathrm{Pr} = \frac{\lambda^{D_f}}{\lambda^D} = \lambda^{-c_F} \tag{2}$$

where $D$ is the dimension of the field (1 for a timeserie, 2 for a spatial field) and $c_F = D - D_f$ is called the fractal codimension of the field.

It is clear that $D_f$ (and $c_F$) depend on the threshold that is used. Several thresholds and corresponding values of $D_f$ are hence required to characterize the field.

In the results section, we have added $D_f$ in Table 4. and we also discuss it in the text.

In terms of fractal dimension $D_f$, it can be seen that for all events, both in space and time, the radar QPE has the most discontinuous binary precipitation field due to its smaller values of $D_f$. COSMO simulations are characterized by larger values of $D_f$ indicating a wider coverage of precipitation and less convoluted precipitation fields and timeseries. It is interesting to notice that the two-moment scheme gives values of $D_f$ that are closer to those of QPE, which indicates that it is better at simulating small-scale variations in the temporal and spatial occurence of precipitation.

For the last part (timeseries of multifractal parameters), it is true that we consider only $\alpha$ and $C_1$, which are the two main parameters of the multifractal framework, and explain the scaling behaviour accross all scales. We have made this more explicit in the text by writing "universal multifractal parameters" instead of simply "multifractal parameters".

4. *Why is the new analysis in part 1 performed on the direct fields - despit $H > 0.5$ for the spatial analysis but on the fluctuation fields in the second part?*

   Thank you for this remark. Indeed, we haven't noticed this inconsistency. Initially, we didn't consider fluctuations of the fields for practical reasons because they are quite expensive to compute, since our dataset is huge. We have thus repeated this analysis by using the fluctuations of the fields when $H > 0.5$. Unfortunately, this seems to break the observed spatial structure of MF parameters. This can be explained by the fact that taking the fluctuations of the fields is only a very simplistic method to dealing with non-conservation. The proper way of dealing with this issue would be to perform a fractional integration in the Fourier space, which would allow a much more smoother correction (the correction will be proportionnal to $H$), which would preserve the spatial structure of the data. Moreover, working with fluctuations will not help when $H < 0$. Ultimately, in order to be consistant we have decided to treat all zones in the same way, just like we treated all timesteps in the same way in the second part of our work, and to consider the original fields instead of their fluctuations. This is now justified in the text:

   Even though some areas are characterized by values of $H > 0.5$ (non-conservative fields), it was decided to treat all areas in a consistant way, by working on the original fields instead of the fluctuations. Indeed, it was observed that using fluctuations for areas where $H > 0.5$, was causing important discontinuities in the spatial structure of MF parameters. Indeed, using fluctuations is only a crude way to address the non-conservativity of the fields. A proper correction using fractional integration, would allow for a much smoother correction, since it is proportional to $H$, but is computationaly intractable because of the very large dataset that is used.

5. *Why are both, Fig. 7 and Fig. 8, needed? Are they not redundant, both providing the same scaling insights?*

   Fig. 7 and 8. are complimentary and they imply different moments, the spectral density is related to the second-moment, while the TM analysis is computed from many moments. However in Fig. 8, for sake of simplicity, we only show the TM analysis with $q = 1.5$. Indeed, we forgot to add the mention that $q = 1.5$, so this has been added to the caption of Fig. 8. Since $\alpha$ and $C_1$ are widely discussed in the paper, we thought it be relevant to show at least one TM analysis, since the slope of the best fit-line in the TM analysis at moment $q$ is equal to $K(q)$. We have made this more clear by changing the first lines of the corresponding explanation in the text:

   Figure 8 shows the trace-moment (TM) analysis in time and space of the three events for $q = 1.5$. The value of $K(q = 1.5)$ is the slope of the best-fit lines. Repeating the TM analysis for various values of $q$ allows to characterize $K(q)$ and to estimate $\alpha$ and $C_1$.

6. *p.21 —l 2: in general, the error of the parameters is not necessarily the same as the significance of the model. In how far can the discrepancies between the spatial and temporal analysis be used to estimate the systematic error of the parameters.*

   Generally, discrepancies between the spatial and temporal analysis cannot systematically be related to estimation errors in the MF parameters because they can also occur when a simple

space/time advection model is not a realistic model for the considered precipitation system, for example when precipitation is not frontal or when strong orographic effects occur, which is often the case in the Alps.

**2.2 Technical corrections**

1. *- p13, l 31 + p 15, l 12, 13: The qualifier "much" does not seem appropriate and could be left out.*

   These qualifiers have been removed according to your suggestions.

2. *- Fig. 3: This is a very busy figure with lots of information. I think a different color scale, which only shows blue and red for high correlation coefficients and lighter/whiter colors in between would make the figure easier to grasp. Also, it could be helpful not to show (or hatch or replace by gray squares) correlations that disagree in sign between the spatial and temporal analysis, thus only showing robust correlations. Also, it would be helpful to visually separate the blocks with descriptors and multifractal parameters.*

   The figure has been changed according to your suggestions, and the caption has been updated.

3. *- Fig. 7: It would be helpful to add fit lines for the COSMO data as well - it is hard to follow the qualitative discussion about whether the 1- or 2-moment schemes scales more like the radar by having to guess where the lines might lie.*

   Fit lines are not really relevant in this case, since no single scaling regime can be observed for the model precipitation intensities. The idea of this plot is more to show how the radar spectral density seems to be more or less linear while the ones of the model are clearly curved.

**3 Anonymous Referee #2**

We thank the anonymous referee #2 for his helpful comments and review and for giving us a chance to publish our work. We have performed the required changes, which are listed below.

**3.1 Minor comments**

1. *There are still issues with grammar, especially comma usage, that make the text hard to follow in places. For example, on Pg. 13, "...two-dimensional geophysical process and in time, we consider..." is confusing without a comma before "and". Moreover, when introducing a sentence with "For x", a comma is needed after "x". Another example is on Pg. 14, Line 4, where a comma is needed before "follows".*

   The synthax and grammar of the paper have been improved and many commas have been added where necessary. The paper has also been proofread by a native speaker who confirmed the modifications and added some.

2. *Throughout the paper, there are many erroneous subsection numbers that should be removed.*

   Unfortunately this issue appeared when preparing the reviewed paper version (with blue fonts) using a macro. We apologize for not seing this when submitting the reviewed paper. This has of course been fixed now.

3. *Throughout the paper, there are many erroneous line breaks within sentences that need to be removed.*

   This issue has been fixed.

4. *Periods are missing at the end of several figure captions.*

   All necessary periods have been added.

5. *The added figures in Appendix A and Appendix B are nice; however, I would suggest just included them in the main text unless you add text surrounding them in the corresponding appendix.*

   We have decided to move the text associated with the corresponding figures to the appendix.

6. *Please review the reference – there are formatting inconsistencies.*

   Inconsistencies in the references have been checked for and should be fixed now.

7. *Pg. 1, Line 1: Reword – "allows to characterize" is awkward.*

   This has been changed to ("allows" was removed from the sentence):

   The framework of universal multifractals (UM) characterizes the spatio-temporal

8. *Pg. 2 Line 34: Change "simulations of" to "simulated".*

   Fixed.

9. *Table 1: Use "Wind speed"; add a space between units to avoid confusion.*

   Fixed.

10. *Pg. 5, Line 2: Should be "These data are".*

    Fixed.

11. *Pg. 5, Line 3: Need to add "the" before "beginning".*

    Fixed.

12. *Pg. 6, Line 17: QPE already defined.*

    The definition has been removed.

13. *Fig. 2: Enlarge to make it easier to read.*

    The figure is now double column, which makes it significantly larger.

14. *Pg. 11, Line 11: Move to end of prior paragraph.*

    Fixed.

15. *Pg. 13, Line 24: Change to "second-order".*

    Fixed.

16. *Fig. 3: The caption is incorrect – there is no left and right panel.* Changed "left" and "right" to "top" and "bottom.

17. *Pg. 15, Line 15: Consider replacing "spectacular" with a more descriptive term.*

    We replaced "spectacular" by "marked".

18. *Pg. 16, Line 1: Define ANOVA and use capital letters.*

We removed the word "anova" from the description of the MANOVA and replaced it by: MANOVA (multivariate analysis of variance).

19. *Pg. 16, Line 2: Change to "tests".* Fixed.

20. *Pg. 18, Line 25: Change "that" to "which".* Fixed.

21. *Pg. 18, Lines 28-30: The wording is awkward; reword.* We have rewritten the sentence as:

This comparison requires to run the COSMO model at the radar temporal resolution (5 minutes) in a very expensive setup (two-moments scheme)

which should be easier to read and understand.

22. *Pg. 20, Line 20: Change to "large-scale".* Fixed.

23. *Pg. 20, Line 25: Use the variables from the table instead of just H.*

Unfortunately, we were not totally sure to understand this point. We have removed the "s" in "multifractal parameters $H$", in order to make it more clear that we compare one single parameter $H$ but for several events and data types.

24. *Pg. 20, Line 32: Use "one-moment scheme".*

Fixed.

25. *Pg. 20, Line 33: You are comparing QPE between the radar and model, not the radar QPE with the scheme itself; reword.*

This has been reworded as:

In space, the trend is not as obvious and the match between the radar QPE and the precipitation intensities simulated with the one-moment scheme seems better.

26. *Pg. 24: Line 2: Use "south".*

Fixed.

27. *Section 5.4: The section is a bit confusing because you start off saying that you will focus on the third event; however, there are two paragraphs on the first and second. Either omit the discussion of the first and second events or reword/reorganize the beginning of the section.*

Maybe this part was confusing, because there are no paragraphs on the first and second events. We have tried to make this more clear by rewriting the beginning.

To study the timeseries of multifractal parameters, the focus is put on the third (convective) event which shows the largest temporal variability. It was observed that the conclusions drawn for the third event in terms of discrepancies between radar and model multifractal parameters can be generalized to all events.

This sentence is then followed by a detailed analysis of all phases of event 3 only. The last paragraph generalizes the conclusions to the other two events as announced at the beginning.

28. *Pg. 20, Line 20: Change to "large-scale".*

Fixed.

29. *Pg. 24, Line 29: This should not be a paragraph on its own.*

    This has been fixed.

30. *Pg. 25, Lines 1-2: The sentence is awkward; reword.*

    The sentence has been reworded to try to make it more clear and explicit:

    Using a mobile window would make the discrepancies in multifractal parameters depend much more on the small-scale structures of simulated precipitation intensities, since it would strongly reduce the effect of misplacements of the simulated precipitation systems.

31. *Figure B3: Place figure title below color bar so that it is clear to the reader what the colors represent. For potential vorticity, use units of 10-5 K m2 kg-1 s-1 to remove the large number of zeroes in the colorbar.*

    We have fixed the figure according to your suggestions.